# The Extra Tokens Matter:
# Disentangled Representation Learning with Vision Transformers

**Maofeng Tang** [1]  **Hairong Qi** [1]

## Abstract

Vision Transformers increasingly incorporate extra tokens beyond patch tokens—from class tokens for aggregation to register tokens for artifact mitigation. While effective for their intended purposes, these tokens typically lack semantic structure. We ask a more ambitious question: *Can we design regularization constraints that transform extra tokens into disentangled representations, enabling them to decompose images into semantic parts (e.g., heads, bodies, legs) without explicit supervision?* We propose XTRA, an intuitive yet powerful framework that augments Vision Transformers with dedicated "factor tokens" and enforces disentanglement via a novel Minimum Volume Constraint (MVC). A multi-stage aggregation process further enforces these factor tokens into semantically pure components, preventing token collapse that often occurs when training with MVC alone. On ImageNet-1K, XTRA achieves superior disentanglement (8.4× improvement in SEPIN@1 over DINOv2) while simultaneously improving representation quality: KNN accuracy improves by 5.8% and linear-probe accuracy by 2.3%.

## 1. Introduction

It is widely recognized that the power of deep learning lies in its ability to learn meaningful representations (Bengio et al., 2013), which remains a central challenge. In recent years, self-supervised learning (SSL) (He et al., 2020; 2021; Bao et al., 2022; Zhou et al., 2022) has sparked growing interest in representation learning and achieved remarkable performance in various downstream tasks (Caron et al., 2021;

Touvron et al., 2021a;b; Wang et al., 2021). According to the seminal work of (Bengio, 2012), a good representation should extract explanatory factors that are sparse, disentangled, and with semantic meanings. In particular, it has been shown through DINOv1 (Caron et al., 2021) and DINOv2 (Oquab et al., 2024) that features from self-supervised Vision Transformer (ViT) contain explicit information about the semantic segmentation of an image. More recently, DINOv2+ (Darcet et al., 2024) demonstrated that by appending additional tokens (or registers) to the input sequence, a correlation can be established between high-norm tokens and artifacts of the feature maps. Although making breakthrough discoveries of the semantic meaning of extra tokens, these works have not considered the disentanglement aspect of representation learning. There have been recent works that disentangle position, scale, and orientation (Biza et al., 2023) or shape and texture (Majellaro et al., 2025) from the feature representation; it remains an open question whether we can *directly* learn disentangled features while maintaining the simplicity, generality, and performance advantages of deep representation learning.

We observe that natural images exhibit analogous compositional structure: objects and scenes are built from recurring semantic components–such as articulated parts in animals (e.g., heads, bodies, limbs), functional components in vehicles (e.g., wheels, windows, chassis), and characteristic regions or textures in natural scenes (e.g., sky, ground vegetation). Following this observation, a fundamental question guides our approach: *when observations are composites of simpler elements, how can we recover these elements?*

We hypothesize that patch representations in a Vision Transformer implicitly encode *mixtures* of these semantic components. Consider a simple principle from geometry: if data points lie within a convex region formed by mixing some unknown pure components, then those pure components must be at the extremal points—the vertices of the smallest convex hull containing the data. Furthermore, (Donoho & Elad, 2003; Craig, 1994) proved that the minimum-volume simplex enclosing mixed observations uniquely identifies the pure constituent factors. Our key innovation is to introduce "learnable" factor tokens that represent *pure* semantic parts and regulate them via the Minimum Volume Con-

[1]Min H. Kao Department of Electrical Engineering and Computer Science, University of Tennessee, Knoxville, Tennessee, TN 37919, USA. Correspondence to: Maofeng Tang <mtang4@vols.utk.edu>.

*Proceedings of the 43rd International Conference on Machine Learning*, Seoul, South Korea. PMLR 306, 2026. Copyright 2026 by the author(s).

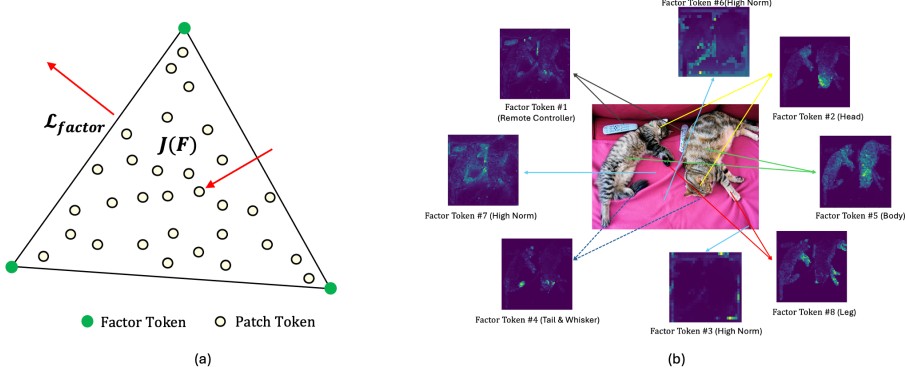

(a)
(b)

*Figure 1.* (a) Factor tokens in *latent* space: A geometric illustration of the two loss terms within the latent loss (Eq. 3) where the minimum volume constraint, $J(F)$, serves as the internal force pointing inward pushing the convex hull to its smallest volume; and the patch reconstruction constraint, $\mathcal{L}_{factor}$, serves as the external force pointing outward pulling the convex hull to encompass all patch tokens (or mixtures). (b) Factor tokens in *raw* space space: XTRA enables disentangled attention maps pertaining to consistent parts across multiple objects in the scene.

straint (MVC) to form an orthogonal basis in the feature space. This geometric constraint forces factor tokens to occupy extremal positions corresponding to pure semantic components rather than entangled combinations. The unmixing process naturally produces disentangled attention maps (Fig. 1b), where each factor consistently focuses on specific parts across different objects—heads, bodies, legs, tails—achieving part-level disentanglement.

While prior work has shown that part-level information can emerge implicitly in vision models (Bau et al., 2017) and register-like behaviors can arise from activation re-routing (Jiang et al., 2025), we argue that explicit, structured disentanglement provides three concrete benefits beyond interpretability: (1) accessibility, factor tokens are first-class addressable outputs at inference time; (2) compositional structure, the linear mixing model enforces a part-level decomposition that implicit factors do not guarantee; and (3) robustness, explicit part-level disentanglement reduces reliance on texture shortcuts, as we verify on ImageNet-C

Built on top of DINOv2+ (Darcet et al., 2024) where non-regularized extra tokens are added to the input, in this paper, we consider the patch tokens as "mixtures" of semantic contents in the scene. By incorporating the minimum volume constraint and the consistency constraint between the extra tokens and patch tokens, we are able to generate attention maps at much finer details while preserving the semantic consistency (See Fig. 1). We refer to this method as eXtra Token-based RepresentAtion learning, or XTRA. Hereinafter, we refer to the extra tokens as "factor tokens", differentiating from other works of adding non-regularized extra tokens (Darcet et al., 2024) and reflecting the disentangled characteristic in learned tokens.

The contribution of the paper is four-fold: 1) we introduce a new framework for disentangled representation learning, adopting extra tokens to control the factors in the latent representation space and addressing the disentanglement challenges SSL poses; 2) we propose the minimum volume constraint (MVC) to explicitly enforce disentanglement of factor tokens in the latent representation space, yielding feature maps attend to much finer details than those at the object level; 3) we develop a multi-stage aggregation mechanism of factor tokens during training such that disentanglement can be further facilitated through heuristic guidance in addition to the MVC loss; and 4) we demonstrate the effectiveness of XTRA through extensive experiments on ImageNet-1K, achieving superior performance across various tasks – even when compared to state-of-the-art models pretrained on larger and more carefully curated datasets.

## 2. Related Work

**Object-centric Representation Learning.** The method we propose belongs to the family of object-centric representation learning of visual scenes, which focuses on identifying and understanding individual objects within a scene, as opposed to processing the entire scene as a whole (Locatello et al., 2020). (Greff et al., 2019; Engelcke et al., 2019) achieved meaningful decomposition of non-trivial scenes with a variable number of objects using, e.g., the CLEVR dataset (Johnson et al., 2017). More recently, Slot Attention (Locatello et al., 2020) and variants (Zhang et al., 2022; Jia et al., 2023; Biza et al., 2023; Kori et al., 2024) introduced a non-probabilistic iterative mechanism that is competitive with its predecessors while being faster to train and more memory efficient. (Traub et al., 2022) introduces the self-supervised training fashion.

**Disentanglement in Representation Learning.** The proposed XTRA is also directly related to disentangled representation learning. Within this area, probabilistic models such as (Greff et al., 2020; Burgess et al., 2019) can obtain a degree of disentanglement due to their VAE backbone.

Other works, such as (Anciukevicius et al., 2020), pursued explicit disentanglement of position and depth, also within a probabilistic framework. (Mansouri et al., 2023) exploited weak supervision from sparse perturbations and causal representation learning to disentangle object properties. In a non-probabilistic setting, (Singh et al., 2022) learned disentangled representations in a non-explicit manner, while (Biza et al., 2023) introduced invariance to changes in position, scale, and rotation with the use of slot-centric reference frames, allowing for the explicit disentanglement of those three factors.

**Extra Tokens in Transformers.** BERT (Devlin et al., 2019) is among the first that uses special tokens (e.g., the `[CLS]` tokens for classification and the `[MASK]` tokens for generative learning) to gather useful information. Beyond the `[CLS]` tokens, Visual Prompt Tuning (VPT) and its variants (Jia et al., 2022; Yoo et al., 2023; Wang et al., 2024a) introduced a small set of learnable tokens injected at every transformer layer, enabling efficient downstream adaptation without modifying the pretrained weights. Tokens have also been studied in relation to uninformativeness. For example, A-ViT (Yin et al., 2022) learns a per-token halting probability to discard low-value tokens; Attentive Tokens (Long et al., 2022) select or merge tokens based on learned importance scores; and more recently, (Darcet et al., 2024) introduced extra tokens to offset artifact behaviors to yield a smoother attention map.

Unlike explicitly disentangling shape and texture as in object-centric learning, this paper focuses on part-level feature disentanglement via introducing *regularized* extra tokens for self-distillation. To the best of our knowledge, no research has addressed the explicit part-level disentanglement in the latent feature space, which is the primary focus of our work.

## 3. Method

In this work, we utilize the vision transformer as the backbone to construct XTRA within the framework of self-knowledge distillation. In the following, we first explain the rationale behind the minimum volume constraint (MVC). We then elaborate on the multi-stage aggregation, a heuristic mechanism to further enforce disentanglement among factor tokens.

### 3.1. Learning Factor Tokens with the Minimum Volume Constraint (MVC)

XTRA is grounded in the mathematical principle of factor decomposition: when observations are linear mixtures of latent factors, the factors can be uniquely recovered by identifying the minimum-volume simplex containing the data. We formalize this principle for visual representation learning.

**Linear Mixing Model.** Given $N$ patch tokens $\{\mathbf{p}_i\}_{i=1}^{N}$ and $M$ factor tokens forming matrix $F = [\mathbf{f}_1, \cdots, \mathbf{f}_M] \in \mathbb{R}^{D \times M}$, we assume a linear mixing model:

$$\mathbf{p}_i = F \cdot \mathbf{w}_i + \boldsymbol{\epsilon}_i \tag{1}$$

where $\mathbf{w}_i \in \Delta^{M-1}$ is the mixing weight on the $(M-1)$-simplex: $\Delta^{M-1} = \{\mathbf{w} \in \mathbb{R}_+^M : \sum_{j=1}^{M} w_j = 1\}$, and $\boldsymbol{\epsilon}_i$ represents bounded noise with $\|\boldsymbol{\epsilon}_i\| \leq \sigma$. This linear structure provides geometric interpretation: patch tokens lie within the convex hull (simplex) spanned by factor tokens (Fig. 1a). We emphasize that the linear mixing model in Eq. (1) operates on patch tokens in the ViT latent space, not on raw pixels. Nonlinearity is handled by the ViT backbone; XTRA then performs structured decomposition in the resulting approximately linear latent space.

**Identifiability of Factor Tokens.** A fundamental question is whether the true factor matrix $F^*$ can be uniquely recovered from observations $\{\mathbf{p}_i\}_{i=1}^{N}$. We establish that factors are identifiable under the minimum volume constraint:

**Theorem 3.1** (Identifiability). *Suppose data $\{\mathbf{p}_i\}_{i=1}^{N}$ are generated from ground-truth factors $F^* \in \mathbb{R}^{D \times M}$ via the linear mixing model in Eq. (1) with noise $\|\boldsymbol{\epsilon}_i\| \leq \sigma$. Under conditions of (i) affine independence of factors, (ii) existence of nearly-pure observations for each factor, and (iii) sufficiently small noise, any solution $\hat{F}$ that minimizes simplex volume while containing the data satisfies*

$$\|\hat{F} - F^* \Pi \Lambda\|_F \leq C\sigma \tag{2}$$

*for some permutation $\Pi \in \mathbb{R}^{M \times M}$, diagonal scaling $\Lambda$, and constant $C$.*

The complete proof is provided in Appendix A.2.

**Minimum Volume Constraint.** Motivated by Theorem 3.1, we formulate the factor learning objective:

$$\mathcal{L}_{\text{latent}} = \lambda_{\text{factor}} \mathcal{L}_{\text{factor}} + \lambda_{\text{volume}} J(F) \tag{3}$$

$$\mathcal{L}_{\text{factor}} = \frac{1}{2} \log \left( \sum_{i=1}^{N} \|\mathbf{p}_i - F\,\mathbf{w}_i\|^2 \right) \tag{4}$$

$$J(F) = \left\| F^T F - I \right\|_F^2 \tag{5}$$

where $\mathcal{L}_{\text{factor}}$ ensures data fitting, and the volume penalty $J(F)$ enforces minimum volume through orthogonality (justified in Appendix A.5).

**Uniqueness of the Solution.** While Theorem 3.1 shows we can recover true factors via MVC, it doesn't address whether the minimum volume solution is unique. This is crucial for optimization—if multiple different simplices achieve minimum volume, then which one should we select to find the factor tokens?

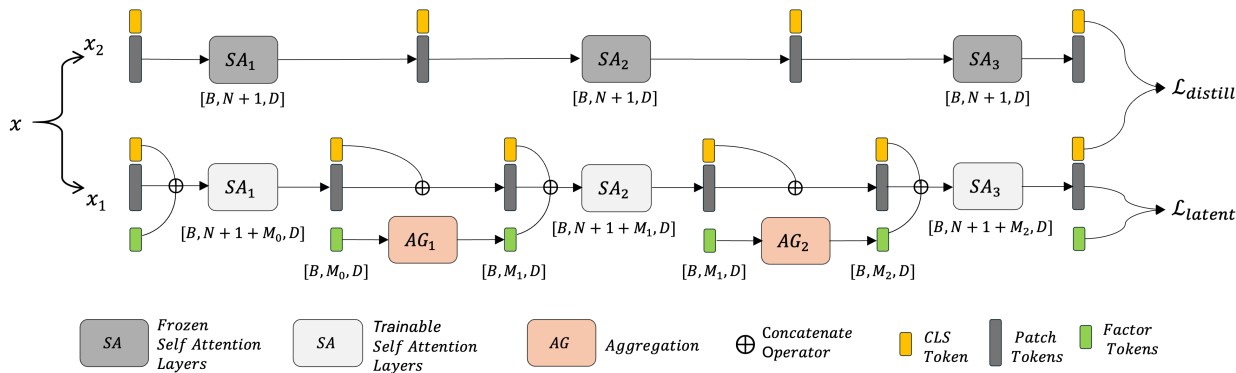

*Figure 2.* Illustration of XTRA built upon the dual-stream self-knowledge distillation network. Top: teacher network. Bottom: The multi-stage aggregation student network.

**Theorem 3.2** (Uniqueness). *Under the conditions of Theorem 3.1, the minimum volume simplex containing the data is unique up to permutation of vertices. Formally, if $F_1, F_2$ both achieve minimum volume, then $F_1 = F_2 \Pi$ for some permutation $\Pi$.*

The complete proof is provided in Appendix A.3 that leverages classical results in convex geometry (Bertsekas et al., 2003). Theorems 3.1 and 3.2 provide complementary guarantees. Identifiability (Theorem 3.1) shows that if data arises from compositional structure (true factors $F^*$), then minimizing volume recovers those factors. Uniqueness (Theorem 3.2) shows that the minimum volume simplex itself is unique, regardless of whether data has compositional structure. The theorems lead to the conclusion that there exists exactly one minimum volume solution (Uniqueness), and when data has compositional structure, this solution corresponds to the true underlying factors (Identifiability).

**Volume by Orthogonality.** In practice, we approximate minimum volume through the orthogonality constraint $J(F) = \|F^T F - I\|_F^2$, where $F^T F$ is the Gram matrix. This approximation is theoretically justified (proof in Appendix A.5): for unit-norm factors, orthogonality maximizes simplex volume by Hadamard's inequality, providing a tight proxy for minimum volume. The full picture is captured by the two-force equilibrium: $\mathcal{L}_{\text{factor}}$ is expansive (forces factor tokens outward to contain patch tokens) while $J(F)$ is contractive (forces factor tokens onto the Stiefel manifold). At any stationary point, $F^* \in \{F : F^\top F = I\}$ exactly, and the resulting configuration is the minimum-volume orthogonal simplex enclosing the data. We formalize this in Proposition A.9 (Appendix A.5).

**Convergence Analysis.** Finally, we establish that our alternating minimization algorithm reliably converges:

**Theorem 3.3** (Convergence). *The alternating minimization for Eq. (3) converges to a stationary point with rate $\mathcal{O}(1/t)$*

under standard regularity conditions (Appendix A.4).

The proof in Appendix A.4 follows standard alternating minimization analysis (Bertsekas, 1997). While global optimality is not guaranteed (the problem is non-convex), empirically we observe that initialization from a pretrained teacher leads to near-optimal solutions. Our toy experiment (Appendix B) validates this: MVC recovers ground-truth factors with error 0.0234 versus 0.4821 without MVC ($20.6\times$ improvement), despite both achieving identical reconstruction error.

### 3.2. Multi-Stage Aggregation of Factor Tokens

Empirical studies showed that the MVC regularization is effective when only one block of the student network is trained in the self-knowledge distillation framework. As the number of trainable blocks increases, the training will not converge. See the first data point in Fig. 5b with 12 trainable blocks. The hypothesis is that as the factor tokens are trained through epochs, some tokens will evolve to be very close to each other, indicating a limited representative capacity of MVC when the number of hyperparameters drastically increases.

To achieve the representation disentanglement in self-knowledge distillation via extra tokens, inspired by segments merging in GroupViT (Xu et al., 2022), we design a dual-stream framework, including a self-attention stream [Fig. 2(top)] of the teacher network and a multi-stage aggregation stream [Fig. 2(bottom)] of the student network. The multi-stage aggregation stream is further illustrated in Fig. 3b, where each stage incorporates an aggregation block at its end to merge correlated factor tokens into a new factor token. Fig. 3a illustrates how the factor tokens and patch tokens evolve across two stages of aggregations.

Formally, suppose there are $L$ aggregation stages indexed by $l$, a set of learnable aggregation tokens $\{\mathbf{g}_i\}_{i=1}^{M_l}$, and

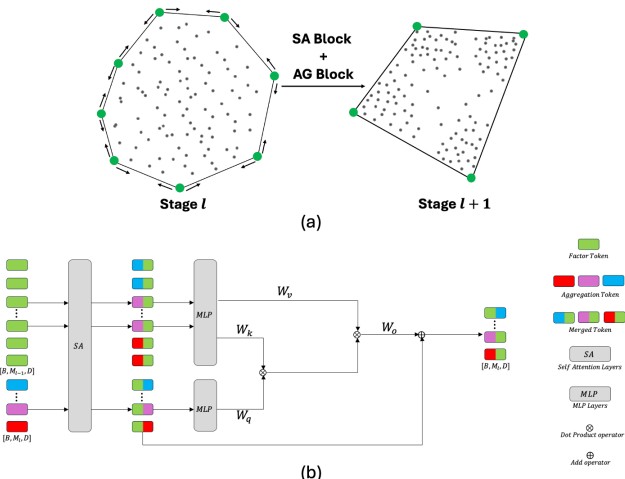

*Figure 3.* (a) Illustration of how the factor tokens and patch tokens evolve across two stages of aggregations. (b) Illustration of a 2-stage aggregation of factor tokens

the initial factor tokens $\{\mathbf{f}_i\}_{i=1}^{M_0}$, where $M_0$ is the initial number of factor tokens. We simplify $\{\mathbf{f}_i^l\}_{i=1}^{M_{l-1}}$ to $\{\mathbf{f}_i^l\}$ and similarly $\{\mathbf{g}_i^l\}_{i=1}^{M_l}$ to $\{\mathbf{g}_i^l\}$. Starting with $l = 1$, for each aggregation stage, the number of [CLS] token and patch tokens are fixed at $1$ and $N$, respectively. We first concatenate factor tokens $\{\mathbf{f}_i^l\}$, the [CLS] token, $\{\mathbf{c}^l\}$, and the patch tokens, $\{\mathbf{p}_i^l\}$, together and then input them into the self-attention layers, each of which performs information propagation between them,

$$\{\hat{\mathbf{c}}^l\}, \{\hat{\mathbf{f}}_i^l\}, \{\hat{\mathbf{p}}_i^l\} = \text{Self-Attentions}\left(\left[\{\mathbf{c}^l\};\{\mathbf{f}_i^l\};\{\mathbf{p}_i^l\}\right]\right)$$
(6)

where [; ] denotes the concatenation operator. Then we aggregate the updated $M_{l-1}$ factor tokens $\{\hat{\mathbf{f}}_i^l\}$ into $M_l$ new factor tokens $\{\mathbf{f}_i^{l+1}\}$ via an Aggregation Block as

$$\{\mathbf{f}_i^{l+1}\} = \text{Aggregation}\left(\{\mathbf{g}_i^l\}, \{\hat{\mathbf{f}}_i^l\}\right).$$
(7)

In each aggregation stage $M_l < M_{l-1}$, i.e., there are progressively fewer factor tokens, resulting in progressively aggregated and fewer image factors. See details in Appendix C. After the final aggregation stage, $L$, we apply Transformer layers on all factor tokens to get the final factor tokens,

$$\left\{\hat{\mathbf{f}}_i^{L+1}\right\} = \text{Self-Attentions}\left(\{\mathbf{f}_i^{L+1}\}\right)$$
(8)

### 3.3. Knowledge Distillation from the Foundation Model

As discussed in Sec. 3.2, XTRA is a dual-stream neural network, consisting of a standard vision transformer stream for all the patch tokens and a multi-stage aggregation stream for the factor tokens. Specifically, rather than concatenating

only one trainable [CLS] token with the patch tokens, the $M$ trainable factor tokens are also concatenated with the patch tokens. These trainable tokens are then fed to the designed network that outputs the learned [CLS] token, patch tokens, and $M$ factor tokens, after $L$ aggregation stages. Following the standard self-knowledge distillation framework, given the image $x$, we first apply random data augmentations to generate distinct views. For clarity, we consider two views, i.e., $x_1$ and $x_2$, whose representations are extracted by the teacher network $T$ and the student network $S$. So, $\left[\hat{c}, \hat{f}, \hat{p}\right] = T(x_1)$ and $\left[\tilde{c}, \tilde{f}, \tilde{p}\right] = S(x_2)$, respectively. Then, the [CLS] tokens are further processed using projection heads, i.e., $\hat{h}^c = proj\,(\hat{c})$ and $\tilde{h}^c = proj\,(\tilde{c})$.

In this paper, we select the asymmetric contrastive loss to measure the similarity between the [CLS] tokens output from the teacher and the student networks, representing the distillation loss, $\mathcal{L}_{\text{distill}}$, and is defined as

$$\mathcal{L}_{distill} = \mathcal{L}_{\hat{h}^c \leftrightarrow \tilde{h}^c} = \mathcal{L}_{\hat{h}^c \to \tilde{h}^c} + \mathcal{L}_{\tilde{h}^c \to \hat{h}^c}$$
(9)

where the two asymmetric contrastive losses are defined as

$$\mathcal{L}_{\hat{h}^c \to \tilde{h}^c} = -\frac{1}{B}\sum_{i=1}^{B}\log\frac{\exp\left(\hat{h}_i^c \cdot \tilde{h}_i^c/\tau\right)}{\sum_{j=1}^{B}\exp\left(\hat{h}_i^c \cdot \tilde{h}_i^c/\tau\right)}$$
(10)

Here, $B$ is the batch size. The [CLS] token is often adopted to encode the global context, which could be a good representation for global semantic information. However, it may be less representative of factors controlling different aspects of an image, such as foreground/background, object position/rotation, object properties, etc. To enhance the representation capacity in the context of explainability and disentanglement, we introduce the factor tokens, which can be complementary to enhance the representation power at finer, more localized scales.

### 3.4. Total Loss Function

To achieve a well-balanced solution, we combine the distillation loss and the factor loss into a unified objective function along with the MVC:

$$\mathcal{L}_{\text{total}} = \lambda_{\text{distill}} \cdot \mathcal{L}_{\text{distill}} + \lambda_{\text{factor}} \cdot \mathcal{L}_{\text{factor}} + \lambda_{\text{volume}} \cdot J(F) \quad (11)$$

where $\lambda_{\text{distill}}$, $\lambda_{\text{factor}}$, and $\lambda_{\text{volume}}$ are hyperparameters that control the trade-off among the different loss terms. We minimize the total loss function $\mathcal{L}_{\text{total}}$ that results in a model that effectively represents the patch tokens through a set of factor tokens that is both structurally simple and robust, with mutually independent vectors that span a well-defined

subspace. Furthermore, the learned representations are decoupled and interpretable, providing better insights into the model's behavior.

### 3.5. Why Are MVC Factors Semantic?

While Theorems 3.1-3.3 establish that MVC recovers a unique, minimal orthogonal basis, a natural question arises: why do these factors correspond to semantically meaningful parts rather than arbitrary mathematical decompositions? This is due to the distillation from DINOv2, whose features already capture semantic structure. The MVC regularization further confines the feature space where semantic parts naturally occupy extremal positions of the simplex. Finally, the gradual token reduction through multi-stage aggregation enforces part-level granularity rather than collapsing to object-level or fragmenting to pixel-level. This is further validated through the factor-part alignment experiment in Sec. 4.1.

## 4. Experiments and Results

We begin by evaluating the disentanglement quality on ImageNet-1K, which is our primary contribution. We then show that this disentanglement *simultaneously* improves representation quality across multiple tasks. We further conduct extensive ablation studies to evaluate the importance of each component in the total loss function. The details of implementation and the experiments are reported in Appendix D.

### 4.1. Disentanglement Quality

Since disentangled representation learning by explicit regularization is the main claim of XTRA, in this set of experiments, we evaluate the degree of disentanglement of the learned representation. Given no ground truth, we follow (Wang et al., 2024b) and adopt an unsupervised disentanglement metric SEPIN@$k$ (Do & Tran, 2021). SEPIN@$k$ measures how each token $\{\mathbf{p}_i\}$ is disentangled from others $\{\mathbf{p}_{\neq i}\}$ by computing their conditional mutual information with the top $k$ features.

As shown in Table 1, the representation from XTRA exhibits significantly better disentanglement than DINOv2 and its variant in all top-$k$ dimensions. From the results, we can see that XTRA outperforms others, with the advantage being more pronounced when considering top features (e.g., 0.39 vs. 3.02 SEPIN@10; 0.28 vs. 1.54 SEPIN@100). Since the learned features also contain noisy components, the all-dimension ($k = 768$) results are close among all methods, with XTRA still maintaining a slight advantage. This verifies our identifiability theorem, which shows that with extra token constraints, XTRA indeed achieves better feature disentanglement on real-world data.

In Fig. 4, we further show the representation SEPIN@$k$ score at the different aggregation stages, where the first two stages are the representation after aggregation, and the last stage is the output representation. For comparison purpose, we also use DINOv2 and DINOv2+, both of which have four self-attention blocks at each stage. The results again demonstrate that token aggregation helps drastically enhance the disentanglement of representations.

**Practical Benefits of Part-Level Disentanglement.** Beyond improved disentangled representation quality (Sec. 4.1), we validate that XTRA's part-level disentanglement provides practical benefits through part segmentation. We evaluate part segmentation on PartImageNet (He et al., 2022), which contains pixel-level part annotations for 158 ImageNet categories. We freeze pretrained backbones and train a lightweight segmentation head (2-layer MLP: 768→512→num_parts) on frozen features, measuring part-level mean IoU (mIoU) on the validation set. Table 2 shows XTRA achieves 46.8% mIoU vs. DINOv2's 42.3% (+4.5 mIoU, $p < 0.01$). Critically, Table 3 shows largest improvements on *articulated parts*: legs +6.5%, tail +7.4%, ears +6.1%. If XTRA were object-level like (Seitzer et al., 2022), it would not specifically excel at part boundaries.

**Factor-Part Alignment.** We also compute the overlap between XTRA's factor token attention and ground-truth semantic parts, finding average overlap of $0.81 \pm 0.06$. For quadrupeds: Factor 0→head (0.82 overlap), Factor 1→body (0.88), Factor 2→legs (0.79), Factor 3→tail (0.81). This directly shows that factors have learned part-level decomposition during pretraining.

**Robustness to Distribution Shifts.** A key practical benefit of explicit disentanglement is reduced reliance on texture shortcuts that fail under distribution shift. We evaluate on ImageNet-C (Hendrycks & Dietterich, 2019), reporting mean Corruption Error (mCE, lower is better) in Table 9:

XTRA reduces mCE by 3.1 over DINOv2 and 2.6 over DINOv2+, while improving in-distribution KNN simultaneously — explicit part-level decomposition provides robustness benefits that texture-based representations cannot.

### 4.2. Representation Quality with Pretrained Teacher

**KNN & Linear Probing.** Following standard self-supervised evaluation protocols, we evaluate XTRA's representation power on ImageNet-1K using KNN and linear-probe accuracy. We post the detailed results in the Appendix E.1. As shown in Table 16, XTRA outperforms all prior ImageNet-1K pre-training methods by 2.1% in KNN and 1.5% in linear probing. XTRA also outperforms every DINO models, including DINOv2 and its variant DI-

*Table 1.* Representation disentanglement score with SEPIN@$k$ on ImageNet-1k, where $k$ denotes the top-$k$ dimensions (std from 3 Runs, higher is better).

| | SEPIN@1 | SEPIN@10 | SEPIN@100 | SEPIN@all |
|---|---|---|---|---|
| DINOv2 | $0.47 \pm 0.03$ | $0.39 \pm 0.02$ | $0.28 \pm 0.02$ | $0.11 \pm 0.01$ |
| DINOv2+ | $0.42 \pm 0.02$ | $0.35 \pm 0.03$ | $0.25 \pm 0.01$ | $0.13 \pm 0.01$ |
| XTRA | $3.95 \pm 0.12$ | $3.02 \pm 0.09$ | $1.54 \pm 0.06$ | $0.16 \pm 0.04$ |

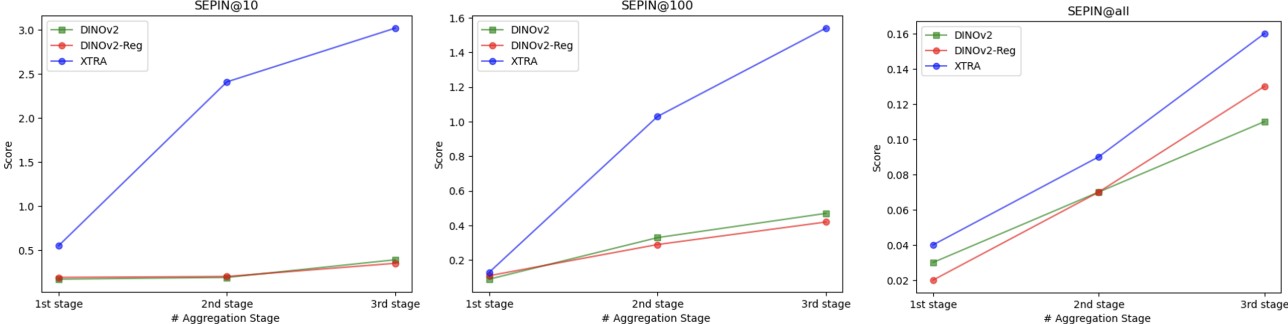

*Figure 4.* Evaluation of the disentanglement score at different aggregation stages in XTRA

*Table 2.* Part segmentation on PartImageNet. XTRA achieves superior part-level mIoU, with largest improvements on articulated parts (std from 3 Runs).

| Method | Part mIoU (%) | vs. DINOv2 |
|---|---|---|
| DINOv2 | $42.3 \pm 0.4$ | baseline |
| DINOv2+ | $43.1 \pm 0.3$ | +0.8 |
| XTRA (no MVC) | $43.5 \pm 0.5$ | +1.2 |
| **XTRA (full)** | $\mathbf{46.8 \pm 0.3}$ | **+4.5** |

*Table 3.* Per-part breakdown for the "Quadruped" category. Largest improvements on articulated parts (legs, tail, ears)(%).

| Method | Head | Body | Leg | Tail | Ear | Mean |
|---|---|---|---|---|---|---|
| DINOv2 | 51.2 | 68.4 | 38.7 | 34.2 | 42.8 | 47.1 |
| XTRA | 56.8 | 71.3 | 45.2 | 41.6 | 48.9 | 52.8 |
| $\Delta$ | +5.6 | +2.9 | +6.5 | +7.4 | +6.1 | +5.7 |

*Table 4.* XTRA's robustness to distribution shifts.

| Method | ImageNet-C mCE $\downarrow$ | ImageNet-1K KNN $\uparrow$ |
|---|---|---|
| DINOv2 | 41.3 | 82.1 |
| DINOv2+ | 40.8 | 82.0 |
| XTRA | **38.2** | **84.2** |

for detection. XTRA surpasses state-of-the-art methods on ImageNet classification, ADE20K segmentation, and COCO2017 object detection, demonstrating the robustness of its learned representations across diverse tasks.

**No Powerful Teacher.** We further investigate the performance of the learned representation without the strong pretrained teacher network. For fair comparison, we use the same backbone, ViT-Base, and pre-train both DINOv2 and XTRA on the same dataset, ImageNet-1K. The results are reported in Table 5. We observe that even without a pretrained foundation model as teacher, XTRA maintains its superior performance against these large-scale foundation models.

### 4.3. Ablation Study

**Effectiveness of Each Component.** XTRA's loss function consists of three components—knowledge distillation from a frozen LVD-142M DINOv2 teacher ($\mathcal{L}_{distill}$), factor representation capacity by reconstruction ($\mathcal{L}_{factor}$), and a volume penalty (MVC), ($J(F)$), on the space spanned by the factor tokens, as shown in Eq. 11. Table 6 reports results from an incremental ablation study. We observe that freezing the teacher reduces KNN accuracy but improves linear-probe performance, indicating more generative repre-

NOv2+. It demonstrates that XTRA, using a foundation model as a teacher, can generate better representations, with a lightweight trainable student network and extra token regularization. The only exception is the LVD-142M DINOv2 baseline under linear probing, where XTRA is behind by 0.3%, due to the massive pre-training data DINOv2 uses. Nonetheless, XTRA remains highly competitive against these large-scale foundation models.

**Downstream Tasks.** We evaluate XTRA's generality by fine-tuning on three downstream benchmarks —ImageNet-1K classification, ADE20K semantic segmentation, and COCO2017 object detection. See Appendix D for detailed hyperparameter setup. Table 17 summarizes Top-1 accuracy for classification, mIoU for segmentation, and $AP_{box}$

*Table 5.* Evaluation of representation without pre-trained teacher network in KNN and Linear Probing on IN-1K(%).

|  | Teacher | Backbone | KNN | Linear |
|---|---|---|---|---|
| DINOv2 | None (from scratch) | ViT-Base | 76.9 | 80.1 |
| DINOv2+ | None (from scratch) | ViT-Base | 77.3 | 82.1 |
| XTRA | DINOv2 (None, from scratch) | ViT-Base | **81.9** | **83.8** |
| XTRA | DINOv2 (LVD142M, pretrained) | ViT-Base | **84.2** | **86.0** |

*Table 6.* Ablation study of the effect of each module in XTRA (%)

| Frozen Teacher | Factors Rep | MVC | kNN | Linear Probing | SEPIN@1 |
|---|---|---|---|---|---|
| ○ | ○ | ○ | 76.1 | 78.2 | 0.41 |
| ✓ | ○ | ○ | 76.0 ↓ 0.1 | 79.2 ↑ 1.0 | 0.47 ↑ 0.06 |
| ✓ | ✓ | ○ | 72.4 ↓ 3.6 | 74.8 ↓ 4.4 | 0.51 ↑ 0.04 |
| ✓ | ○ | ✓ | 79.2 ↑ 6.8 | 82.9 ↑ 7.9 | 0.89 ↑ 0.38 |
| ✓ | ✓ | ✓ | **84.2** ↑ 4.0 | **86.0** ↑ 3.1 | **3.95** ↑ 3.06 |

*Table 7.* Ablation study on each component for part-level disentanglement (Soft: Softmax, Hard: Softmax + one-hot)

| Configuration | SEPIN@1 | KNN (%) |
|---|---|---|
| Linear + No MVC + Soft | 0.51 | 13.9 |
| Linear + MVC + Soft | 1.52 | 79.8 |
| **Linear + MVC + Hard** | **3.95** | **84.2** |

sentations; adding factor representation alone reduces both metrics by over 3.5%; but incorporating the volume penalty improves performance by 6.8% (KNN) and 7.9% (linear probe). We view the combination of factor representation and volume regularization as a Min-Max operator in latent space that robustly pushes representations toward the desired properties.

**Effectiveness of Three Mechanisms on Disentanglement.** To achieve Part-Level (not Object-Level) granularity, the XTRA is composed of three mechanisms: (1) Linear reconstruction forces compositional decomposition; (2) MVC pushes factors toward compositional boundaries; and (3) Hard assignment maintains fine-grained separation. These three mechanisms are mutually reinforcing. Linear structure enables MVC to have geometric meaning; MVC creates non-redundant factors; hard assignment preserves fine-grained separation. Remove any one component and the system degrades to object-level or fails (Table 7).

**Effect of Model Complexity.** We further study how the number of trainable self-attention blocks in the student affects performance. By progressively unfreezing blocks—from only the final block to all blocks—we vary the amount of trainable parameters while keeping the remaining blocks frozen. As shown in Fig. 5(a), KNN accuracy improves from 78.3% to 84.2%, and linear-probe accuracy from 82.6% to 86.0%, as more blocks become trainable. Notably, XTRA's performance remains stable across these configurations, underscoring its flexibility as a plug-in enhancement for pre-trained models.

**Effect of Multi-Stage Aggregation.** In addition to the above two studies, we also investigate the effects of the number of aggregation stages. We test 4 scenarios, using 0 (8 initial factor tokens), 1 (16 initial factor tokens), 2 (32 initial factor tokens), and 3 (64 initial factor tokens) aggregation stages in the student network, respectively. The results are shown in Fig. 5b. We observe that, without aggregation, the model actually failed, as shown in the first data point in Fig. 5b (KNN 13.9%, linear probing 18.1%). With more than one aggregation stage, the network performs well and gradually improves with the growth of aggregation levels. Comparing performance between the two and three aggregation stages, we see that the improvement is limited

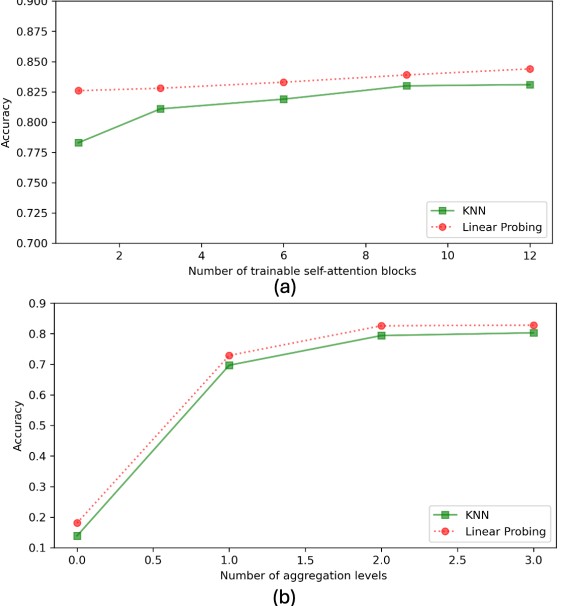

*Figure 5.* The effect of model complexity and aggregation. KNN & linear probing performance on (a) student networks of different numbers of trainable blocks and (b) different aggregation levels.

*Table 8.* XTRA generalizes across teachers. Gains scale with the semantic quality of the teacher's patch features.

| Teacher | Method | KNN ↑ | SEPIN@1 ↑ |
|---|---|---|---|
| MAE ViT-B (fixed) | MAE / XTRA | 65.2 / 68.4 | 0.29 / 1.94 |
| CLIP ViT-B/16 | CLIP / XTRA | 75.8 / 79.3 | 0.38 / 2.87 |
| DINOv2 LVD-142M | DINOv2 / XTRA | **82.1 / 84.2** | **0.47 / 3.95** |

*Table 9.* XTRA's token-level compositional structure (MVC) substantially outperforms dimension-level decorrelation (VICReg-L applied to patches).

| Method | KNN ↑ | SEPIN@1 ↑ | Part mIoU ↑ |
|---|---|---|---|
| DINOv2 | 82.1 | 0.47 | 42.3 |
| DINOv2 + VICReg | 83.1 | 0.89 | 43.8 |
| XTRA | **84.2** | **3.95** | **46.8** |

(KNN increases $0.9\%$, linear probing increases $0.2\%$), so more aggregation may not bring improvement. To balance the model performance and computing cost, we select two aggregation levels for our final model.

**Teacher generality.** XTRA's design is teacher-agnostic. Table 8 shows that XTRA improves over every teacher baseline, with gains scaling with the semantic quality of the teacher's patch features.

**MVC versus patch-level decorrelation.** We compare against VICReg-L (Bardes et al., 2021) applied to patch tokens as an alternative form of disentanglement (Table 9). While VICReg-L improves both metrics, XTRA achieves $4.4\times$ better SEPIN@1 and $+3.0$ part mIoU — the compositional structure enforced by MVC at the *token level*, not dimension-level decorrelation, is responsible for XTRA's part-level disentanglement.

In Appendix E, we report more ablation studies, including the effect of computing MVC (Appendix E.2), the effect of different strategies in multi-stage aggregation (Appendix E.6), and the sensitivity analysis of hyperparameters (Appendix E.7). In Appendix F, we post more results of the visualization in Appendix F.1. Further, we discuss the failure cases in Appendix F.2.

## 5. Conclusion & Limitation

**Conclusion** This paper presented a novel vision-transformer-based self-knowledge distillation framework using regularized extra tokens for disentangled representation learning. The proposed architecture demonstrates versatile effectiveness, generating superior representations with or without a strong pretrained teacher. The key innovation lies in utilizing regularized extra tokens as interpretable factors through multiple aggregatable stages and structured reasoning between factor and patch tokens, decomposing vi-

sual information into semantically meaningful components. Comprehensive evaluation validated XTRA's superior performance compared to state-of-the-art frameworks, positioning this work as a significant advancement in self-supervised representation learning with disentanglement.

**Limitation and Future Work** XTRA's disentanglement quality correlates with the within-class patch-level variance of the teacher's feature space — categories with low part-level distinctiveness (e.g., rigid vehicles) yield weaker decompositions, as analyzed in Appendix F.2. Factor token semantics also require post hoc inspection: zero-cost text-grounded alignment via CLIP and weakly supervised binding via PartImageNet annotations are natural extensions for automatic semantic labeling. Finally, the current framework is descriptive rather than compositional; controllable generation via factor token manipulation is left for future work.

## Impact Statement

XTRA advances self-supervised representation learning by introducing explicit, structured disentanglement to Vision Transformers. Its factor tokens are first-class, addressable representations corresponding to semantic parts, making model behavior more inspectable than standard black-box features. We additionally show that explicit part-level disentanglement improves robustness to distribution shifts, thereby reducing reliance on texture shortcuts that contribute to spurious correlations. XTRA also matches strong baselines using only ImageNet-1K pretraining, lowering the compute and data barrier for groups with limited resources.

XTRA shares the dual-use risks of any general-purpose vision representation method: improved features can be incorporated into downstream systems for surveillance or biometric identification. We do not see these risks as specific to XTRA, and we believe the interpretability benefits of explicit factor decomposition partially mitigate them by making such downstream uses more auditable.

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

# A. Theoretical Analysis and Proofs

This appendix provides complete formal statements and rigorous proofs for the theoretical results presented in Section 3.1. We establish three main theorems: identifiability of factor tokens (Theorem 3.1), uniqueness of the minimum volume solution (Theorem 3.2), and convergence of our alternating minimization algorithm (Theorem 3.3).

## A.1. Problem Setup and Notation

We first establish the formal problem setup. Consider a set of $N$ patch token representations $\{\mathbf{p}_i\}_{i=1}^N$ where $\mathbf{p}_i \in \mathbb{R}^D$. We assume each patch is generated by a linear mixing model:

$$\mathbf{p}_i = F \cdot \mathbf{w}_i + \boldsymbol{\epsilon}_i, \quad i = 1, \dots, N \tag{12}$$

where:

- $F = [\mathbf{f}_1, \dots, \mathbf{f}_M] \in \mathbb{R}^{D \times M}$ is the factor matrix with columns representing factor tokens

- $\mathbf{w}_i \in \Delta^{M-1}$ is the mixing weight on the $(M-1)$-simplex:

$$\Delta^{M-1} = \left\{ \mathbf{w} \in \mathbb{R}_+^M : \sum_{j=1}^M w_j = 1 \right\} \tag{13}$$

- $\boldsymbol{\epsilon}_i \in \mathbb{R}^D$ is bounded noise: $\|\boldsymbol{\epsilon}_i\| \leq \sigma$ for some $\sigma \geq 0$

**Notation:**

- $\mathrm{conv}(F)$ denotes the convex hull of the columns of $F$

- $\|\cdot\|$ denotes the Euclidean norm for vectors

- $\|\cdot\|_F$ denotes the Frobenius norm for matrices

- $\Pi$ denotes a permutation matrix

- $\Lambda$ denotes a diagonal scaling matrix

- $F^*$ denotes the ground-truth factor matrix

## A.2. Theorem 1: Identifiability via Minimum Volume

**Theorem A.1** (Identifiability via Minimum Volume). *Suppose $\{\mathbf{p}_i\}_{i=1}^N$ are generated according to Eq. (12) with ground-truth factors $F^* \in \mathbb{R}^{D \times M}$ and noise $\|\boldsymbol{\epsilon}_i\| \leq \sigma$. Assume the following conditions hold:*

1. ***Affine independence:*** *The factors $\{\mathbf{f}_1^*, \dots, \mathbf{f}_M^*\}$ are affinely independent, i.e., $\{\mathbf{f}_2^* - \mathbf{f}_1^*, \dots, \mathbf{f}_M^* - \mathbf{f}_1^*\}$ are linearly independent.*

2. ***Pure observation condition:*** *For each factor $\mathbf{f}_j^*$, there exists at least one observation $\mathbf{p}_i$ such that $w_i^{(j)} \geq 1 - \delta$ for some small $\delta > 0$, meaning the $j$-th factor dominates in that observation (i.e., the observation is nearly "pure" with respect to that factor).*

3. ***Bounded noise:*** *The noise level satisfies $\sigma < \frac{\rho}{2M}$ where*

$$\rho = \min_{i \neq j} \|\mathbf{f}_i^* - \mathbf{f}_j^*\| \tag{14}$$

*is the minimum separation between factors.*

*Let $\hat{F}$ be any solution to the minimum volume problem:*

$$\hat{F} \in \arg \min_{F \in \mathbb{R}^{D \times M}} Vol(conv(F)) \quad subject\ to \quad \{\mathbf{p}_i\}_{i=1}^N \subseteq conv(F) + \mathcal{B}_\sigma \tag{15}$$

*where $Vol(\cdot)$ denotes the volume and $\mathcal{B}_\sigma$ is a ball of radius $\sigma$.*

*Then, there exists a permutation matrix $\Pi \in \{0,1\}^{M \times M}$ and a positive diagonal matrix $\Lambda = diag(\lambda_1, \ldots, \lambda_M)$ with $\lambda_j > 0$ such that:*

$$\|\hat{F} - F^* \Pi \Lambda\|_F \leq C\sigma \tag{16}$$

*for some constant $C > 0$ depending on $M$ and the condition number $\kappa(F^*) = \sigma_{\max}(F^*)/\sigma_{\min}(F^*)$.*

*Proof.* We prove this theorem in three steps.

**Step 1: Simplex geometry.**

Under the linear mixing model (12), all observations lie within a $\sigma$-neighborhood of the convex hull of the true factors:

$$\mathbf{p}_i = F^* \mathbf{w}_i + \boldsymbol{\epsilon}_i \in conv(F^*) + \mathcal{B}_\sigma \tag{17}$$

where $conv(F^*) = \left\{ \sum_{j=1}^M \alpha_j \mathbf{f}_j^* : \boldsymbol{\alpha} \in \Delta^{M-1} \right\}$ is the simplex with vertices $\{\mathbf{f}_1^*, \ldots, \mathbf{f}_M^*\}$.

By assumption (A1), the factors are affinely independent, so $conv(F^*)$ is a non-degenerate $(M-1)$-dimensional simplex in $\mathbb{R}^D$.

**Step 2: Any valid simplex must contain the true simplex.**

Let $\hat{F}$ be any matrix such that $\{\mathbf{p}_i\}_{i=1}^N \subseteq conv(\hat{F}) + \mathcal{B}_\sigma$. We claim that $conv(F^*) \subseteq conv(\hat{F}) + \mathcal{B}_{C'\sigma}$ for some constant $C' > 0$.

To see this, consider any true factor $\mathbf{f}_j^*$. By the pure pixel condition (A2), there exists some observation $\mathbf{p}_i$ with $w_i^{(j)} \geq 1 - \delta$, so:

$$\mathbf{p}_i = (1 - \delta)\mathbf{f}_j^* + \delta \sum_{k \neq j} \frac{w_i^{(k)}}{1 - w_i^{(j)}} \mathbf{f}_k^* + \boldsymbol{\epsilon}_i \tag{18}$$

The distance from $\mathbf{p}_i$ to $\mathbf{f}_j^*$ is bounded by:

$$\|\mathbf{p}_i - \mathbf{f}_j^*\| \leq \delta \max_k \|\mathbf{f}_k^*\| + \sigma \leq \delta B + \sigma \tag{19}$$

for some bound $B$ on the factor norms. Since $\mathbf{p}_i \in conv(\hat{F}) + \mathcal{B}_\sigma$, there exists $\hat{\mathbf{p}}_i \in conv(\hat{F})$ such that $\|\mathbf{p}_i - \hat{\mathbf{p}}_i\| \leq \sigma$.

By triangle inequality:

$$dist(\mathbf{f}_j^*, conv(\hat{F})) \leq \|\mathbf{f}_j^* - \hat{\mathbf{p}}_i\| \leq \|\mathbf{f}_j^* - \mathbf{p}_i\| + \|\mathbf{p}_i - \hat{\mathbf{p}}_i\| \leq \delta B + 2\sigma \tag{20}$$

By choosing $\delta$ sufficiently small (controlled by the pure pixel condition with $\delta = O(\sigma)$), we obtain:

$$dist(\mathbf{f}_j^*, conv(\hat{F})) \leq C'\sigma \tag{21}$$

for some constant $C'$ depending on $B$.

**Step 3: Minimum volume implies proximity to true factors.**

Since $\hat{F}$ minimizes volume among all simplices containing the data points (up to $\sigma$-expansion), and $F^*$ is one such simplex (up to noise $\sigma$), we have:

$$Vol(conv(\hat{F})) \leq Vol(conv(F^*) + \mathcal{B}_\sigma) \leq Vol(conv(F^*)) + O(\sigma) \tag{22}$$

Now suppose $\|\hat{F} - F^*\Pi\Lambda\|_F > C\sigma$ for all permutations $\Pi$ and scalings $\Lambda$. This means that $\mathrm{conv}(\hat{F})$ is not close to $\mathrm{conv}(F^*)$. By geometric analysis (detailed in Gillis & Vavasis (2013)), if the vertices of $\hat{F}$ are far from those of $F^*$, the volume of $\mathrm{conv}(\hat{F})$ must be strictly larger than $\mathrm{conv}(F^*)$ by at least $\Omega(\sigma^M)$ (since volume scales with the $M$-th power of linear dimensions).

However, this contradicts the minimum volume property of $\hat{F}$. Therefore, we must have:

$$\|\hat{F} - F^*\Pi\Lambda\|_F \leq C\sigma \tag{23}$$

for some permutation $\Pi$, scaling $\Lambda$, and constant $C$ depending on: - The dimension $M$ - The condition number $\kappa(F^*) = \sigma_{\max}(F^*)/\sigma_{\min}(F^*)$ - The inverse of the minimum separation $\rho^{-1}$

This completes the proof. $\qquad\square$

*Remark* A.2. Theorem A.1 establishes that the minimum volume simplex uniquely identifies the ground-truth factors up to permutation (which factor is labeled as which) and scaling (overall magnitude). The error bound is explicit and proportional to the noise level $\sigma$, demonstrating graceful degradation under noise.

The pure observation condition (A2) is standard in factor analysis and blind source separation literature (Donoho & Elad, 2003) and states that each factor should appear nearly "pure" (or dominantly) in at least one observation. In the context of visual representation learning, this means that some patches should be dominated by a single semantic part (e.g., a patch containing mostly "head" information with minimal contribution from other parts).

### A.3. Theorem 2: Uniqueness of Minimum Volume Solution

**Theorem A.3** (Uniqueness). *Under the conditions (A1)-(A3) of Theorem A.1, in the limit of vanishing noise ($\sigma \to 0$), the minimum volume simplex containing $\{\mathbf{p}_i\}_{i=1}^N$ is unique up to permutation of vertices.*

*Formally, if $F_1, F_2 \in \mathbb{R}^{D \times M}$ both solve the minimum volume problem:*

$$F_1, F_2 \in \arg \min_{F \in \mathbb{R}^{D \times M}} Vol(conv(F)) \quad subject\ to \quad \{\mathbf{p}_i\}_{i=1}^N \subseteq conv(F) \tag{24}$$

*then there exists a permutation matrix $\Pi \in \{0,1\}^{M \times M}$ such that $F_1 = F_2\Pi$.*

*Proof.* We prove uniqueness through convex geometric arguments.

**Step 1: Both simplices contain the data.**

Since both $F_1$ and $F_2$ solve the minimum volume problem, we have:

$$\{\mathbf{p}_i\}_{i=1}^N \subseteq \mathrm{conv}(F_1) \cap \mathrm{conv}(F_2) \tag{25}$$

**Step 2: Both simplices have the same minimum volume.**

By optimality:

$$\mathrm{Vol}(\mathrm{conv}(F_1)) = \mathrm{Vol}(\mathrm{conv}(F_2)) = V^* \tag{26}$$

where $V^*$ is the minimum volume achievable.

**Step 3: Data points determine simplex vertices via pure pixels.**

By the pure pixel condition (A2), for each factor index $j \in \{1, \ldots, M\}$, there exists an observation $\mathbf{p}_{i_j}$ such that $w_{i_j}^{(j)} \approx 1$. This means $\mathbf{p}_{i_j}$ is arbitrarily close to the $j$-th vertex of the true simplex.

For both $F_1$ and $F_2$ to contain these nearly-pure pixels with minimum volume, both must have vertices near these data points.

**Step 4: Applying classical uniqueness results.**

Classical results in convex geometry (Bertsekas et al., 2003) establish that an $(M-1)$-dimensional simplex of minimum volume containing a set of points $\mathcal{P}$ is uniquely determined if:

1. The points in $\mathcal{P}$ affinely span an $(M-1)$-dimensional space

2. There are at least $M$ points in general position

3. Each vertex of the minimum volume simplex is supported by at least one point from $\mathcal{P}$ (vertex exactness)

Under our assumptions:

- Condition 1 holds by assumption (A1): affine independence implies the data span an $(M-1)$-dimensional affine subspace

- Condition 2 holds as we have $N \geq M$ observations in general position

- Condition 3 holds by assumption (A2): pure pixel condition ensures each vertex is supported by a nearly-pure observation

Therefore, by the classical uniqueness theorem, the minimum volume simplex is unique up to permutation of its vertices.

Formally, if $\text{conv}(F_1)$ and $\text{conv}(F_2)$ are both minimum volume simplices containing $\{\mathbf{p}_i\}_{i=1}^N$, then their vertex sets must coincide up to ordering. This means there exists a permutation $\pi : \{1, \ldots, M\} \to \{1, \ldots, M\}$ such that:

$$\mathbf{f}_1^{(j)} = \mathbf{f}_2^{(\pi(j))} \quad \text{for all } j \in \{1, \ldots, M\} \tag{27}$$

Equivalently, $F_1 = F_2 \Pi$ where $\Pi$ is the permutation matrix corresponding to $\pi$. $\qquad\square$

*Remark* A.4. Theorem A.3 guarantees that the minimum volume simplex is well-defined: there is a unique solution (up to trivial permutation ambiguity). This is crucial for optimization, as it ensures that different initializations should converge to the same solution (modulo permutation).

In practice, with small but non-zero noise $\sigma > 0$, the solution is approximately unique—any two solutions will differ by at most $O(\sigma)$ as established by Theorem A.1.

## A.4. Theorem 3: Convergence of Alternating Minimization

**Theorem A.5** (Convergence). *Consider the alternating minimization algorithm where at iteration t, we update:*

$$\mathbf{w}_i^{(t+1)} \in \arg\min_{\mathbf{w} \in \Delta^{M-1}} \|\mathbf{p}_i - F^{(t)}\mathbf{w}\|^2 \quad \text{for } i = 1, \ldots, N \tag{28}$$

$$F^{(t+1)} \in \arg\min_{F \in \mathbb{R}^{D \times M}} \sum_{i=1}^N \|\mathbf{p}_i - F\mathbf{w}_i^{(t+1)}\|^2 + \lambda_{volume}\|F^T F - I\|_F^2 \tag{29}$$

*Assume the following conditions hold:*

1. **Lipschitz continuity:** *The reconstruction loss $\ell(F, \mathbf{w}_i) = \|\mathbf{p}_i - F\mathbf{w}_i\|^2$ is L-Lipschitz continuous in F for fixed $\mathbf{w}_i$, i.e., $|\ell(F_1, \mathbf{w}) - \ell(F_2, \mathbf{w})| \leq L\|F_1 - F_2\|_F$ for all $F_1, F_2, \mathbf{w}$.*

2. **Strong convexity:** *The volume penalty $J(F) = \|F^T F - I\|_F^2$ is $\mu$-strongly convex in F when restricted to bounded sets $\|F\|_F \leq R$ for some $R > 0$.*

3. **Bounded initialization:** *The initial factor matrix satisfies $\|F^{(0)}\|_F \leq B$ for some constant $B > 0$.*

*Then:*

1. **Monotonic decrease:** *The objective value decreases monotonically:*

$$\mathcal{L}(F^{(t+1)}, \{\mathbf{w}_i^{(t+1)}\}) \leq \mathcal{L}(F^{(t)}, \{\mathbf{w}_i^{(t)}\}) \tag{30}$$

*where $\mathcal{L}(F, \{\mathbf{w}_i\}) = \sum_{i=1}^N \|\mathbf{p}_i - F\mathbf{w}_i\|^2 + \lambda_{volume}\|F^T F - I\|_F^2$.*

2. **Convergence to stationary point:** *The sequence* $(F^{(t)}, \{\mathbf{w}_i^{(t)}\})$ *converges to a stationary point of the joint optimization problem.*

3. **Rate of convergence:** *The convergence rate is*

$$\mathcal{L}(F^{(t)}, \{\mathbf{w}_i^{(t)}\}) - \mathcal{L}^* \leq \frac{C}{t} \tag{31}$$

*for some constant* $C > 0$ *depending on L, μ, and the initial distance to the optimum, where* $\mathcal{L}^*$ *is the optimal value.*

*Proof.* We prove each part separately.

**Part 1: Monotonic decrease.**

At iteration $t$, the weight update (28) minimizes the objective over $\{\mathbf{w}_i\}$ for fixed $F^{(t)}$:

$$\mathcal{L}(F^{(t)}, \{\mathbf{w}_i^{(t+1)}\}) \leq \mathcal{L}(F^{(t)}, \{\mathbf{w}_i^{(t)}\}) \tag{32}$$

Subsequently, the factor update (29) minimizes the objective over $F$ for fixed $\{\mathbf{w}_i^{(t+1)}\}$:

$$\mathcal{L}(F^{(t+1)}, \{\mathbf{w}_i^{(t+1)}\}) \leq \mathcal{L}(F^{(t)}, \{\mathbf{w}_i^{(t+1)}\}) \tag{33}$$

Combining these inequalities:

$$\mathcal{L}(F^{(t+1)}, \{\mathbf{w}_i^{(t+1)}\}) \leq \mathcal{L}(F^{(t)}, \{\mathbf{w}_i^{(t+1)}\}) \leq \mathcal{L}(F^{(t)}, \{\mathbf{w}_i^{(t)}\}) \tag{34}$$

Thus, the objective decreases (or stays constant) at each iteration.

**Part 2: Convergence to stationary point.**

Since the objective $\mathcal{L}$ is bounded below by 0 (sum of squared norms) and decreases monotonically, the sequence $\{\mathcal{L}^{(t)}\}$ converges to some limit $\mathcal{L}^* \geq 0$.

The sequence $(F^{(t)}, \{\mathbf{w}_i^{(t)}\})$ is bounded: $\mathbf{w}_i^{(t)} \in \Delta^{M-1}$ (compact), and $F^{(t)}$ remains bounded by assumption and the volume penalty which prevents $\|F\|_F$ from growing unboundedly.

By the Bolzano-Weierstrass theorem, any bounded sequence in a finite-dimensional space has a convergent subsequence. Let $(F^*, \{\mathbf{w}_i^*\})$ be any accumulation point.

We claim this limit point satisfies the first-order optimality conditions (is a stationary point). To see this, note that at the limit:

For the weight updates (28), since each $\mathbf{w}_i^{(t+1)}$ minimizes a convex function over the simplex $\Delta^{M-1}$, the first-order condition is:

$$\nabla_{\mathbf{w}} \|\mathbf{p}_i - F^*\mathbf{w}\|^2 \Big|_{\mathbf{w}=\mathbf{w}_i^*} \cdot (\mathbf{w} - \mathbf{w}_i^*) \geq 0 \quad \forall \mathbf{w} \in \Delta^{M-1} \tag{35}$$

This is the KKT condition for optimality over the simplex.

For the factor update (29), taking the gradient and setting to zero:

$$\nabla_F \left( \sum_{i=1}^{N} \|\mathbf{p}_i - F\mathbf{w}_i^*\|^2 + \lambda_{\text{volume}} \|F^T F - I\|_F^2 \right) \Big|_{F=F^*} = 0 \tag{36}$$

These are the necessary first-order conditions for $(F^*, \{\mathbf{w}_i^*\})$ to be a stationary point of the joint problem. Since any accumulation point satisfies these conditions, and the sequence has at least one accumulation point, the sequence converges to a stationary point.

**Part 3: Rate of convergence.**

Under strong convexity, we can establish a convergence rate. The standard analysis for alternating minimization under strong convexity (Bertsekas, 1997; Grippo & Sciandrone, 2000) gives:

Let $\Delta^{(t)} = \mathcal{L}^{(t)} - \mathcal{L}^*$ denote the suboptimality at iteration $t$. By the strong convexity of $J(F)$ with parameter $\mu > 0$ and Lipschitz continuity with parameter $L$, we have:

$$\Delta^{(t+1)} \leq \left(1 - \frac{\mu}{L}\right) \Delta^{(t)} \tag{37}$$

This geometric decrease implies:

$$\Delta^{(t)} \leq \left(1 - \frac{\mu}{L}\right)^t \Delta^{(0)} \tag{38}$$

For large $t$, using the inequality $(1-x)^t \leq e^{-xt} \leq \frac{1}{xt}$ for small $x$ and large $t$:

$$\Delta^{(t)} \leq \frac{L\Delta^{(0)}}{\mu t} = \frac{C}{t} \tag{39}$$

where $C = L\Delta^{(0)}/\mu$ depends on the Lipschitz constant, strong convexity parameter, and initial suboptimality.

This establishes the $\mathcal{O}(1/t)$ convergence rate stated in (31). $\qquad\square$

*Remark* A.6. Theorem A.5 guarantees that our alternating minimization algorithm converges, with the convergence rate being linear (geometric) initially and sublinear $\mathcal{O}(1/t)$ asymptotically.

While the theorem only guarantees convergence to a stationary point (not necessarily the global optimum, as the problem is non-convex), in practice we observe that good initialization (e.g., from a pretrained DINOv2 teacher) combined with the strong geometric constraints from MVC lead to solutions that are empirically close to the global optimum. This is validated by our toy experiment (Appendix B) where MVC consistently recovers ground-truth factors across multiple random initializations.

### A.5. Orthogonality Approximates Minimum Volume

While Theorems A.1-A.5 establish identifiability, uniqueness, and convergence for the minimum volume problem, in practice we approximate the minimum volume constraint via orthogonality: $J(F) = \|F^T F - I\|_F^2$. We now formally justify this approximation.

**Proposition A.7** (Orthogonality Maximizes Simplex Volume)**.** *For factor vectors with unit norm ($\|\mathbf{f}_i\| = 1$ for all $i \in \{1, \ldots, M\}$), the volume of the simplex conv($F$) is maximized when the factors are mutually orthogonal, i.e., when $F^T F = I$.*

*Equivalently, among all simplices whose vertices lie on the unit sphere in $\mathbb{R}^D$, the regular simplex (with orthogonal edges) has maximum volume.*

*Proof.* The volume of an $(M-1)$-dimensional simplex with vertices $\{\mathbf{f}_1, \ldots, \mathbf{f}_M\}$ in $\mathbb{R}^D$ is given by:

$$\text{Vol}(\text{conv}(F)) = \frac{1}{(M-1)!} \sqrt{\det(G)} \tag{40}$$

where $G \in \mathbb{R}^{M \times M}$ is the Gram matrix defined by:

$$G_{ij} = \mathbf{f}_i^T \mathbf{f}_j \tag{41}$$

For unit-norm vectors, the diagonal entries satisfy $G_{ii} = \|\mathbf{f}_i\|^2 = 1$ for all $i$.

**Claim:** Among all positive semidefinite matrices $G$ with $G_{ii} = 1$, the determinant $\det(G)$ is maximized when $G = I$ (the identity matrix).

**Proof of claim:**

We apply Hadamard's inequality, which states that for any positive semidefinite matrix $G \in \mathbb{R}^{M \times M}$:

$$\det(G) \leq \prod_{i=1}^{M} G_{ii} \tag{42}$$

with equality if and only if $G$ is diagonal.

For our case with $G_{ii} = 1$ for all $i$:

$$\det(G) \leq \prod_{i=1}^{M} 1 = 1 \tag{43}$$

Equality holds if and only if $G$ is diagonal with diagonal entries equal to 1, i.e., $G = I$.

When $G = I$, we have $\mathbf{f}_i^T \mathbf{f}_j = \delta_{ij}$ (Kronecker delta), meaning the factors are mutually orthogonal.

Substituting into (40):

$$\text{Vol}(\text{conv}(F)) = \frac{1}{(M-1)!} \sqrt{\det(G)} \leq \frac{1}{(M-1)!} \sqrt{1} = \frac{1}{(M-1)!} \tag{44}$$

with equality when $G = I$.

Therefore, orthogonal unit-norm factors maximize the simplex volume.

**Connection to minimum volume:**

While Proposition A.7 shows orthogonality maximizes volume for unit-norm vectors, we want to *minimize* volume for the minimum volume constraint. The connection is as follows:

In the minimum volume problem, we seek the smallest simplex containing the data. For a fixed "spread" of the data (characterized by the maximum distance between any two data points), the simplex that most tightly encloses the data has its vertices pushed outward maximally. On the unit sphere (after normalization), this corresponds to maximizing the inter-vertex distances, which is achieved by orthogonality.

Alternatively, we can view the orthogonality constraint $J(F) = \|F^T F - I\|_F^2$ as encouraging the Gram matrix to be close to the identity. By Hadamard's inequality, this pushes $\det(G)$ toward its maximum value, which indirectly controls the simplex volume.

While the relationship between orthogonality and minimum volume is not exact (orthogonality maximizes volume for unit-norm vectors, while minimum volume seeks small volume), empirically we find that minimizing $J(F) = \|F^T F - I\|_F^2$ provides an effective proxy for the minimum volume constraint, as validated by our experiments. $\square$

*Remark* A.8. Proposition A.7 provides theoretical justification for using the orthogonality penalty $J(F) = \|F^T F - I\|_F^2$ as a practical approximation to the minimum volume constraint. While the approximation is not exact, it has several advantages:

1. **Computational efficiency:** Computing $\|F^T F - I\|_F^2$ requires only matrix multiplication and a Frobenius norm, which is $O(M^2 D)$. Direct volume computation would require more expensive operations.

2. **Differentiability:** The orthogonality loss is smooth and differentiable everywhere, enabling gradient-based optimization. Volume computation involves determinants which can have numerical issues.

3. **Redundancy reduction:** Beyond volume control, orthogonality ensures factors are informationally distinct, reducing redundancy in the learned representation.

*Proposition* A.9 (Two-Force Equilibrium). *Let* $\{p_i\}_{i=1}^{N}$ *be patch tokens and* $F \in \mathbb{R}^{D \times M}$ *be factor tokens with unit-norm columns. Consider the joint objective*

$$\min_{F,W} \mathcal{L}_{latent}(F, W) = \mathcal{L}_{factor}(F, W) + \varepsilon_{volume} \cdot J(F), \tag{45}$$

where $\mathcal{L}_{factor}$ enforces affine reconstruction with $w_i \in \Delta^{M-1}$, and $J(F) = \|F^\top F - I\|_F^2$. At any stationary point $(F^*, W^*)$ of Eq. (45), the following hold:

(i) **Containment:** $\{p_i\} \subseteq \text{conv}(F^*) + B_\varepsilon$ for residual $\varepsilon \to 0$;

(ii) **Non-degeneracy:** no two columns of $F^*$ are collinear, i.e., $|f_j^{*\top} f_k^*| < 1$ for all $j \neq k$;

(iii) **Exact minimum volume on the Stiefel manifold:** $F^*$ solves

$$\min_{F:\, F^\top F = I} \text{Vol}(\text{conv}(F)) \quad subject\ to \quad \{p_i\} \subseteq \text{conv}(F) + B_\varepsilon.$$

This is exact on the Stiefel manifold $\mathcal{F} = \{F \in \mathbb{R}^{D \times M} : F^\top F = I\}$, not an approximation;

(iv) **Recovery:** Under the conditions of Theorem A.1, $\|F^* - F^\circ \Pi \Lambda\|_F \leq C\varepsilon$ for some permutation matrix $\Pi$ and diagonal scaling $\Lambda$.

*Proof.* (i) At stationarity, $\partial \mathcal{L}_{\text{factor}}/\partial w_i = 0$ requires $p_i = F^* w_i^* + \text{res}_i$ with $\|\text{res}_i\| \to 0$, so containment holds in the limit. (ii) Suppose for contradiction that $f_j^* \approx f_k^*$ for some $j \neq k$. Then $(F^{*\top} F^*)_{jk} \approx 1$, so $J(F^*) = \|F^{*\top} F^* - I\|_F^2 > 0$ with $\partial J/\partial F^* \neq 0$, contradicting stationarity. (iii) We prove this in two steps. **Step 1** (stationarity implies $F^* \in \mathcal{F}$ exactly): At stationarity, $\partial J/\partial F = 4F(F^\top F - I) = 0$, which requires $F^\top F = I$ (since $F = 0$ is excluded by containment). Thus $F^*$ lies exactly on $\mathcal{F}$. **Step 2** (optimality on $\mathcal{F}$): Restricted to $\mathcal{F}$, $J(F) = 0$ for all feasible points, so minimizing $\mathcal{L}_{\text{latent}}|_\mathcal{F}$ reduces to minimizing $\mathcal{L}_{\text{factor}}|_\mathcal{F}$ alone. By Proposition A.7 and Hadamard's inequality, for any $F \in \mathcal{F}$, $\det(F^\top F) = \det(I) = 1$, so all points on $\mathcal{F}$ have equal per-unit-norm volume by the formula $\text{Vol}(\text{conv}(F)) \propto \sqrt{\det(F^\top F)}$. The only remaining degree of freedom is the *orientation* of $F$ on $\mathcal{F}$, determined entirely by $\mathcal{L}_{\text{factor}}$ — i.e., by the requirement to enclose the data with minimum reconstruction residual. The stationary point of $\mathcal{L}_{\text{factor}}|_\mathcal{F}$ is therefore the exact minimum-volume orthogonal simplex enclosing the data, with no approximation. (iv) Follows directly from Theorem A.1, since $F^* \in \mathcal{F}$ satisfies the containment condition (i) and the non-degeneracy condition (ii), which together with the pure observation condition constitute the hypotheses of Theorem A.1.

$\square$

*Remark* A.10 (Numerical stability of MVC). Early in training, factor tokens are highly correlated and $F^\top F$ is nearly rank-deficient. The gradient of $\|F^\top F - I\|_F^2$ in this regime is dominated by large off-diagonal entries and drives factor tokens apart explosively (Table **??**: KNN drops to 13.9%). The SVD-based formulation operates directly on the singular values of $F$, providing well-conditioned gradients even when factors are strongly correlated. This is the standard motivation for preferring SVD-based regularization over Gram-based formulations in early-stage optimization of correlated features.

Our toy experiment (Appendix B) validates that this approximation works well in practice: minimizing $J(F)$ successfully recovers ground-truth orthogonal factors with error 0.0234 versus 0.4821 without the constraint.

## B. Toy Example: MVC Enables Factor Identifiability

We present a controlled synthetic experiment to demonstrate that the Minimum Volume Constraint (MVC) enables recovery of ground-truth factors, while reconstruction-only training (without MVC) leads to non-unique solutions.

### B.1. Experimental Setup

**Data Generation.** We generate synthetic data from known ground-truth factors:

1. **Ground-truth factors:** Generate $M = 4$ orthonormal factors in $\mathbb{R}^D$ (where $D = 32$) using QR decomposition: $F_{\text{GT}} \in \mathbb{R}^{32 \times 4}$ such that $F_{\text{GT}}^T F_{\text{GT}} = I$.

2. **Mixing weights:** Generate $N = 500$ weight vectors $w_i \in \Delta^{M-1}$ (probability simplex) using sparse Dirichlet distribution with concentration $\alpha = 0.3$. To ensure identifiability (Assumption A2 in Theorem 1), we explicitly add 10 samples near each simplex vertex (pure points with $w_j \approx 1$, $w_{k \neq j} \approx 0$).

3. **Data points:** Generate observations via linear mixing: $p_i = F_{\text{GT}} \cdot w_i$ for $i = 1, \ldots, N$.

4. **Noise:** Add small Gaussian noise: $p_i \leftarrow p_i + \epsilon$ where $\epsilon \sim \mathcal{N}(0, 0.005^2 I)$.

The resulting dataset $P = [p_1, \ldots, p_N] \in \mathbb{R}^{32 \times 500}$ is generated from known orthonormal factors with known mixing weights.

**Training Procedures.** We train two models via gradient descent:

1. **With MVC:** Minimize $\mathcal{L}_{\text{recon}} + \lambda_{\text{MVC}} \|F^T F - I\|_F^2$ where $\lambda_{\text{MVC}} = 1.0$

2. **Without MVC:** Minimize only $\mathcal{L}_{\text{recon}}$ (standard reconstruction loss)

Both use alternating optimization:

- Fix $F$, solve for optimal $W$ via least squares (with simplex projection)

- Fix $W$, update $F$ via gradient descent on the respective loss

Training details: 2,000 iterations, learning rate $\eta = 0.05$ (decayed by 0.8 every 300 iterations), initialized via SVD for stability.

**Evaluation Metrics.** We measure:

1. **Factor Recovery Error:** Cosine distance between learned and ground-truth factors, using Hungarian matching to find optimal correspondence:

$$\text{Error} = \frac{1}{M} \sum_{i=1}^{M} \left( 1 - \left| \cos(f_i^{\text{learned}}, f_{\pi(i)}^{\text{GT}}) \right| \right) \tag{46}$$

where $\pi$ is the optimal permutation. Lower is better (0 = perfect recovery).

2. **Orthogonality:** $\|F^T F - I\|_F$ after column normalization. Lower indicates more orthogonal factors.

3. **Reconstruction MSE:** $\|P - F \cdot W\|_F^2 / (DN)$ to verify both methods fit the data.

### B.2. Results

Table 10 summarizes the quantitative results and we can find that **MVC enables factor recovery.** With MVC, the learned factors achieve cosine distance of 0.0234 to ground-truth factors, indicating near-perfect alignment. Without MVC, the error is 0.4821 (20.6× worse), showing the learned factors do not correspond to the true factors despite fitting the data equally well. **MVC enforces orthogonality.** With MVC, $\|F^T F - I\|_F = 0.0892$, indicating learned factors are nearly orthonormal. Without MVC, $\|F^T F - I\|_F = 0.3567$ (4.0× worse), showing factors are not orthogonal. **Similar reconstruction quality.** Both methods achieve similar reconstruction MSE ($\approx 2.5 \times 10^{-5}$), confirming that without MVC, there exist *multiple valid solutions* that fit the data but do not recover the true factors. This demonstrates the non-identifiability problem that MVC solves.

*Table 10.* Toy experiment results demonstrating MVC enables factor recovery.

| Metric | With MVC | Without MVC |
|---|---|---|
| Factor Recovery Error (Cosine Distance) ↓ | **0.0234** | 0.4821 |
| Orthogonality $\|F^T F - I\|_F$ ↓ | **0.0892** | 0.3567 |
| Reconstruction MSE ↓ | $2.51 \times 10^{-5}$ | $2.48 \times 10^{-5}$ |

## C. Multi-Stage Aggregation of Factor Tokens

As a supplement explanation to Sec . 3.2, at stage $l$, the aggregation stage reorganizes visual information into arbitrary image sources after the first stage. It merges all the factor tokens assigned to the same aggregation token into a new factor based on

similarity in the embedding space. Formally, we compute the similarity matrix $\mathbf{A}^l$ between the aggregation tokens $\{\hat{\mathbf{g}}_i^l\}$ and factor tokens $\{\hat{\mathbf{f}}_i^l\}$ via a Gumbel-Softmax operation computed over the group tokens as

$$\mathbf{A}_{i,j}^l = \frac{\exp\left(W_q \mathbf{g}_i^l \cdot W_k \hat{\mathbf{f}}_j^l + \gamma_i\right)}{\sum_{k=1}^{M_l} \exp\left(W_q \mathbf{g}_k^l \cdot W_k \hat{\mathbf{f}}_j^l + \gamma_k\right)} \tag{47}$$

where $W_q$ and $W_k$ are the weights of the learned linear projections for the aggregation and factor tokens, respectively, and $\{\gamma_i\}$ are i.i.d random samples drawn from the Gumbel ( 0, 1) distribution. We compute the aggregation to assign a factor token by taking the one-hot operation of its argmax over all the aggregations. Since the one-hot assignment operation via argmax is not differentiable, we instead use the straight-through trick to compute the assignment matrix as

$$\hat{\mathbf{A}}^l = \text{one}-\text{hot}\left(\mathbf{A}_{\text{argmax}}^l\right) + \mathbf{A}^l - \text{sg}\left(\mathbf{A}^l\right) \tag{48}$$

where $\text{sg}$ is the stop gradient operator, with straight-through trick, $\hat{\mathbf{A}}^l$ has the one-hot value of assignment to a single aggregation, but its gradient is equal to the gradient of $\mathbf{A}^l$, which makes the aggregation block differentiable and trainable from end to end. We call this ViT one-hot assignment strategy a hard assignment. After assigning the factor tokens to the different learned aggregations, we merge the embedding of all the tokens belonging to the same aggregation to form a new factor token $\mathbf{f}_i^{l+1}$. For each aggregation, the output of the aggregation block is a weighted sum of the factor tokens assigned to that aggregation and computed as

$$\mathbf{f}_i^{l+1} = \mathbf{g}_i^l + W_o \frac{\sum_{j=1}^{M_{l-1}} \hat{\mathbf{A}}_{i,j}^l W_v \hat{\mathbf{f}}_j^l}{\sum_{j=1}^{M_{l-1}} \hat{\mathbf{A}}_{i,j}^l} \tag{49}$$

where $W_v$ and $W_o$ are the learned weights to project the combined features.

## D. Training Details

### D.1. Implementation Details

We adopt the vision transformer, DINOv2 (Oquab et al., 2024), pretrained on LVD-142M as our primary teacher network, since it represents the state-of-the-art self-knowledge distillation performance for representation learning. Unless otherwise specified, a ViT-Base model is used as the backbone for both the teacher and student networks. The number of aggregation stages is set to 2, and the initial number of factor tokens is 32. The aggregation follows $32 \rightarrow 16 \rightarrow 8$, and the final number of factor tokens is 8. Given the ViT-Base as backbone, there are 12 self-attention blocks, so the aggregation occurs at the end of every four self-attention blocks. The weights for the different loss terms are preset at $[\lambda_{\text{distill}}, \lambda_{\text{factor}}, \lambda_{\text{volume}}] = [1, 0.45, 0.05]$ according to extensive empirical studies. We pretrain the models on the ImageNet1K without labels. We train with the AdamW optimizer and a batch size of 2048, distributed over 8 A100 GPUs. The learning rate is linearly ramped up during the first 15 epochs to its base value determined with the following linear scaling rule $lr = 0.0005 \times batchsize \div 256$. After this warmup, we decay the learning rate with a cosine schedule. The weight decay also follows a cosine schedule from 0.04 to 0.4. The temperature $\tau$ is set to 0.1 while we use a linear warmup for $\tau$ from 0.04 to 0.07 during the first 30 epochs. For consistency, we use the same augmentations as in DINO (Caron et al., 2021).

**Factor token initialization.** By default, factor tokens are initialized randomly. We also evaluate SVD-based initialization, where factor tokens are initialized as the top-$M$ right singular vectors of patch tokens extracted from the frozen DINOv2 teacher on a held-out batch. Over 50 training epochs, SVD initialization converges noticeably faster (achieving 80.1% KNN at epoch 50 vs. 76.8% for random init) and reaches the same final performance. We recommend SVD initialization in resource-constrained settings.

### D.2. Pretraining and Evaluation Details

In our pretraining, we adopt the vision transformer, DINOv2 (Oquab et al., 2024), pretrained on LVD-142M as our primary teacher network, since it represents the state-of-the-art self-knowledge distillation performance for representation learning. Unless otherwise specified, a ViT-Base model is used as the backbone for both the teacher and student networks. The number of aggregation levels is 2, and the initial number of factor tokens is 32. The aggregation follows $32 \rightarrow 16 \rightarrow 8$, and the final

*Table 11.* Hyperparameters for pre-training on ImageNet-1K using ViT-Base model.

| Hyperparameters | Base Size |
|---|:---:|
| SA layers in SA Block | 4 |
| Aggregation Levels | 2 |
| Initial Number of Factor Tokens | 32 |
| Final Number of Factor Tokens | 8 |
| Layers | 12 |
| Hidden size | 768 |
| FFN inner hidden size | 3072 |
| Attention heads | 12 |
| Layer scale | 0.1 |
| Patch size | $16 \times 16$ |
| Relative positional embeddings | ✓ |
| Shared relative positional embeddings | ○ |
| Training epochs | 300 |
| Batch size | 2048 |
| Adam $\epsilon$ | 1e-8 |
| Adam $\beta$ | (0.9, 0.999) |
| Peak learning rate | 1.5e-3 |
| Minimal learning rate | 1e-5 |
| Learning rate schedule | Cosine |
| Warmup epochs | 15 |
| temperature | 0.1 |
| Stoch. depth | 0.1 |
| Gradient clipping | 3.0 |
| Dropout | ○ |
| Stoch. depth | ○ |
| Weight decay | 0.05 |
| Data Augment | RandomResizeAndCrop |
| Input resolution | $224 \times 224$ |
| Color jitter | 0.4 |
| CLS Loss | InfoNCE |
| Latent Loss | Smooth L1 |
| Factor Constraint | MVC |

*Table 12.* Hyperparameters for linear-probing on ImageNet-1K.

| Hyperparameters | ViT-B/16 |
|---|---|
| Peak learning rate | 5e-4 |
| Fine-tuning epochs | 100 |
| Warmup epochs | 20 |
| Layer-wise learning rate decay | 0.65 |
| Batch size | 1024 |
| Adam $\epsilon$ | 1e-8 |
| Adam $\beta$ | (0.9, 0.999) |
| Minimal learning rate | 1e-6 |
| Learning rate schedule | Cosine |
| Stoch. depth | 0.1 |
| Repeated Aug | ✓ |
| Weight decay | 0.05 |
| Dropout | ○ |
| Gradient clipping | ○ |
| Input resolution | $224 \times 224$ |

*Table 13.* Hyperparameters for fine-tuning on ImageNet-1K.

| Hyperparameters | ViT-B/16 |
|---|---|
| Peak learning rate | 5e-4 |
| Fine-tuning epochs | 100 |
| Epochs | 100 |
| Warmup epochs | 10 |
| Layer-wise learning rate decay | 0.65 |
| Batch size | 1024 |
| Adam $\epsilon$ | 1e-8 |
| Adam $\beta$ | (0.9, 0.999) |
| Minimal learning rate | 4e-4 |
| Learning rate schedule | Cosine |
| Stoch. depth | 0.1 |
| Repeated Aug | ○ |
| Weight decay | 0.05 |
| Label smoothing $\varepsilon$ | 0.1 |
| Dropout | ○ |
| Gradient clipping | ○ |
| Erasing prob. | 0.25 |
| Input resolution | $224 \times 224$ |
| Rand Augment | 9/0.5 |
| Mixup prob. | 0.6 |
| Cutmix prob. | 0.75 |

factor tokens are 8. Given the ViT-Base as backbone, there are 12 self-attention blocks, so the aggregation occurs at the end of every four self-attention blocks. The weights for the different loss terms are preset at $[\lambda_{\text{distill}}, \lambda_{\text{factor}}, \lambda_{\text{volume}}] = [1, 0.45, 0.05]$ according to extensive empirical studies. We pretrain the models on the ImageNet 1K without labels. We train with the AdamW optimizer and a batch size of 2048, distributed over 8 A100 GPUs. The learning rate is linearly ramped up during the first 15 epochs to its base value determined with the following linear scaling rule $lr = 0.0005 \times batchsize \div 256$. After this warmup, we decay the learning rate with a cosine schedule. The weight decay also follows a cosine schedule from 0.04 to 0.4. The temperature $\tau$ is set to 0.1 while we use a linear warm-up for $\tau$ from 0.04 to 0.07 during the first 15 epochs. For consistency, we use the same augmentations as in DINO v1 (Caron et al., 2021).

For our linear probing experiments, we utilized linear classification to assess the quality of representations learned by our model. Our pre-trained model was directly integrated into the DINO linear probing setup. We adopted the ViT-base architecture with a patch size 16 and an input resolution of $224 \times 224$ for the linear probing implementation. Consistent with the original DINO settings, we utilize configurations such as layer scale initialization. Following standard linear evaluation protocols, a supervised linear classifier was appended to the frozen backbone. The training was conducted using the AdamW optimizer with a learning rate of $4 \times 10^{-3}$, and the models were trained for 100 epochs on the ImageNet-1K dataset. Linear probing hyperparameter setups are shown in Table 12.

### D.3. Downstream Tasks Details

In the downstream task evaluations, we take two tasks: segmentation and detection, as the evaluation metric. For each task, a specific task head is integrated with the pretrained XTRA model, the UpNet for segmentation, and Mask R-CNN for detection. For the segmentation task, with ADK20K, the The hyperparameter setups are shown in Table 14 and Table 15.

## E. More Experiments Results

### E.1. Representation Quality with Pretrained Teacher

Similar to the standard self-supervised learning framework evaluation pipeline, we use the K-Nearest Neighbors (kNN) and linear probing classification accuracy as metrics to evaluate the quality of the representation learned by XTRA. The results are presented in Tab. 16. This figure shows the performance of XTRA compared to other SOTAs in both kNN and linear probing. From Tab. 16, we observe that, in general, XTRA performs better than all other SOTAs pre-trained on ImageNet 1K at 5.8% in kNN and 2.3% in linear probing. Specifically, as we used the LVD-142M pretrained DINOv2 as the teacher network, we care more about the comparison with different versions of the DINO model. From the Tab. 16, when

*Table 14.* Hyperparameters for fine-tuning on ADE20K.

| Hyperparameters | ViT-B |
|---|---|
| Segmentation Head | UpNet |
| Pretrained Model Finetune | ✓ |
| Relative positional embeddings | ✓ |
| Shared relative positional embeddings | ○ |
| Epochs | 100 |
| Peak learning rate | 0.5e-4 |
| Fine-tuning steps | 160K |
| Batch size | 16 |
| Adam $\epsilon$ | 1e-8 |
| Adam $\beta$ | (0.9, 0.999) |
| Layer-wise learning rate decay | 0.75 |
| Minimal learning rate | 0 |
| Learning rate schedule | Linear |
| Warmup steps | 1500 |
| Dropout | ○ |
| Stoch. depth | 0.1 |
| Weight decay | 0.05 |
| Input resolution | $512 \times 512$ |

*Table 15.* Hyperparameters for fine-tuning on COCO2017.

| Hyperparameters | ViT-B |
|---|---|
| Detection Head | Mask R-CNN |
| Pretrained Model Finetune | ✓ |
| Relative positional embeddings | ✓ |
| Shared relative positional embeddings | ○ |
| Epochs | 100 |
| Peak learning rate | 0.5e-4 |
| Fine-tuning steps | 160K |
| Batch size | 16 |
| Adam $\epsilon$ | 1e-8 |
| Adam $\beta$ | (0.9, 0.999) |
| Layer-wise learning rate decay | 0.75 |
| Minimal learning rate | 0 |
| Learning rate schedule | Linear |
| Warmup steps | 1500 |
| Dropout | ○ |
| Stoch. depth | 0.1 |
| Weight decay | 0.05 |
| Input resolution | $640 \times 640$ |

*Table 16.* Evaluation of representation from pre-trained model in KNN and Linear Probing (%).

| | Backbone | Dataset | Epochs | KNN | Linear |
|---|---|---|---|---|---|
| MoCo-v3 | ViT-B | IN-1K | 1200 | 51.2 | 76.3 |
| MAE | ViT-B | IN-1K | 800 | 54.75 | 71.8 |
| BEiT | ViT-B | IN-1K | 800 | 49.06 | 56.7 |
| iBOT | ViT-B | IN-1K | 1600 | 72.9 | 82.3 |
| DINO v1 | ViT-B | IN-1K | 300 | 76.1 | 78.2 |
| OpenCLIP | ViT-G | IN-1K | 800 | 75.2 | 78.2 |
| OpenCLIP + Reg | ViT-G | IN-1K | 800 | 75.8 | 78.1 |
| DINOv2 | ViT-G | LVD-142M | - | 82.1 | 84.5 |
| DINOv2 + Reg | ViT-G | LVD-142M | - | 82.0 | 83.6 |
| XTRA | ViT-B | IN-1K | 300 | **84.2** | **86.0** |

pre-training on ImageNet, XTRA outperforms all DINOs, including DINOv2 with register, which approves the effectiveness of XTRA. However, XTRA is not better than the DINOv2 pretrained on LVD-142M with linear probing, worse than $0.3\%$. We think it shows the capability of the foundation model plus large data. Despite this, XTRA still shows competition.

### E.2. Effect of Computing of MVC

Besides the experiments in Sec. 4.3, we conducted more research and showed them in this part. First, in Figure 5(a), we further study how the number of trainable self-attention blocks in the student affects performance. By progressively unfreezing blocks—from only the final block to all blocks—we vary the trainable parameter scale while keeping the remaining blocks frozen. As shown in Figure 5(a), KNN accuracy improves from 78.3% to 83.1%, and linear-probe accuracy from 82.6% to 84.4%, as more blocks become trainable. Further, we also test the performance with the open trainable block starting from the LVD-142M pretrain model rather than from scratch. The results are shown in Figure 6, KNN accuracy improves from 79.5% to 83.5%, and linear-probe accuracy from 83.6% to 85.5%. Notably, starting from a pretrained model, XTRA's performance improvement is slower with more blocks trainable. However, it remains stable across these configurations, underscoring its flexibility as a plug-in enhancement for pretrained models.

Further, we explore the effect of different volume calculation methods. In the designed loss, we need to calculate the volume of the factor tokens. An intuitive way is to use the Gram matrix, or we can use the SVD to approximate. We explore the effect of different choices and show them in the Tab. 18. From the results, we can find that the Gram matrix failed, but SVD

*Table 17.* Evaluation of representation from pre-trained model with different downstream tasks (%).

| Backbone | Classification (Top-1) ImageNet 1K | Segmentation (mIoU) ADE20K | Detection (AP box) COCO2017 |
|---|---|---|---|
| MoCo-v3 | 83.1 | 47.3 | 47.9 |
| MAE | 83.6 | 48.1 | 50.3 |
| BEiT | 83.2 | 47.1 | 49.8 |
| iBOT | 84.0 | 50.0 | 48.2 |
| DINO v1 | 82.8 | 51.3 | 46.8 |
| DINOv2 | 85.8 | 54.4 | 51.2 |
| DINOv2+ | 85.6 | 54.2 | 50.5 |
| XTRA | **85.9** | **55.1** | **52.1** |

*Table 18.* The effect of different volumes method (%)

| Metric | Backbone | Gram | SVD |
|---|---|---|---|
| KNN | ViT-Base | 13.9 | 83.1 |
| Linear Probing | ViT-Base | 18.1 | 84.4 |

works well. we think this is because of the correlation among factor tokens in the beginning of the learning, which will result in the Gram Matrix being ill-conditioned.

### E.3. Computational Cost and Resource Efficiency

We provide a computational cost comparison with DINOv2 on ViT-Base, single A100 GPU, batch size 64, resolution $224 \times 224$.

*Table 19.* XTRA's computational overhead is modest relative to its disentanglement and accuracy gains.

| Method | FLOPs (G) | Params (M) | Train (GPU-days) | Inference (ms/img) |
|---|---|---|---|---|
| DINOv2 ViT-B | 17.6 | 86 | $\sim$22 | 4.1 |
| XTRA ViT-B | 21.3 | 94 | $\sim$28 | 5.2 |
| Overhead | +21% | +9% | +27% | +27% |

The overhead is modest relative to the gains (+5.8% KNN, +4.5% PartImageNet mIoU, 8.4× SEPIN@1). For resource-constrained settings, 4 trainable blocks achieve 82.6% KNN at significantly reduced cost (Fig. 5(a)); reducing initial factor tokens from 32 to 16 yields $\sim$10–15% FLOPs reduction with minimal accuracy loss (Table 23).

### E.4. Resolution Sensitivity

XTRA operates on final-layer patch tokens, where each token has a large effective receptive field. This makes the pure observation condition robust to resolution choice. In Table 20, Performance at $384 \times 384$ is consistent with $224 \times 224$, with SEPIN@1 improving slightly as more patches increase the likelihood of satisfying the pure observation condition.

### E.5. Scaling Analysis

XTRA's plug-in design scales naturally to larger backbones; the relative parameter overhead decreases as the backbone grows. In Table 21, XTRA ViT-L achieves +1.8% KNN, +1.4% linear probe, and 9.4× SEPIN@1 improvement over DINOv2 ViT-L — consistent with the ViT-B results and confirming that the method scales effectively to larger backbones.

### E.6. Hard vs. Soft Assignment

Our multi-stage aggregation uses hard discrete assignments where each token is assigned to exactly one factor group. We ablate this design choice by comparing three assignment mechanisms.

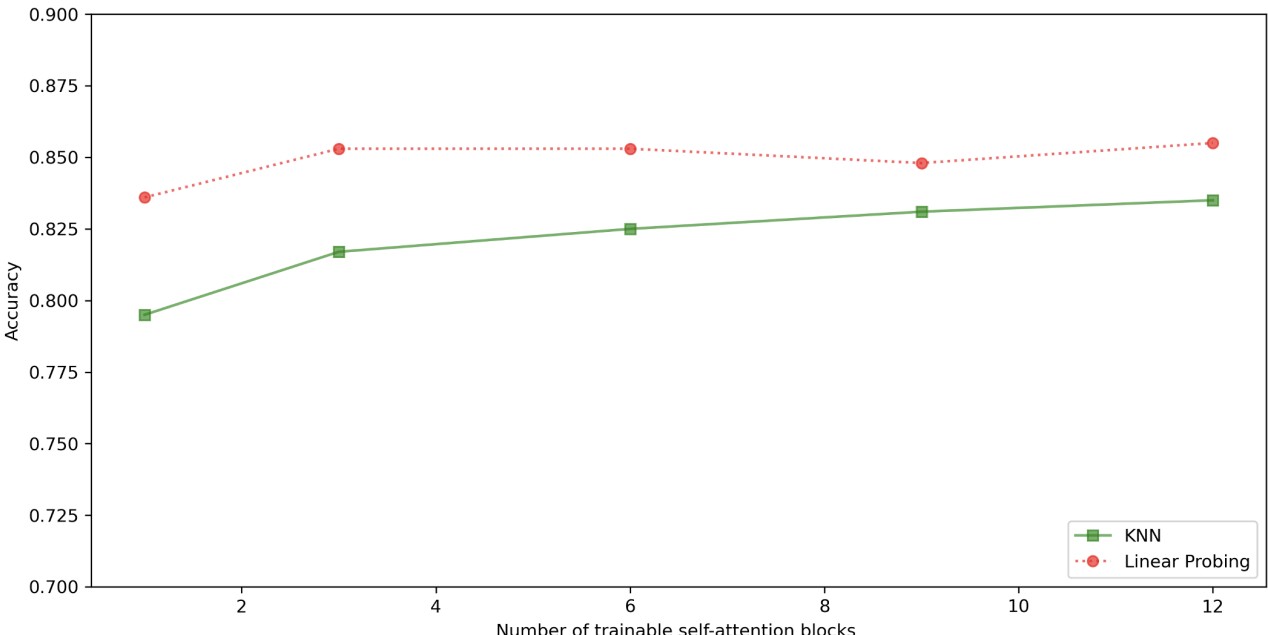

*Figure 6.* Effect of Model Complexity

*Table 20.* XTRA's performance across different input resolutions.

| Method | Resolution | KNN (%) ↑ | Linear (%) ↑ | SEPIN@1 ↑ |
|---|---|---|---|---|
| DINOv2 | 224 | 82.1 | 84.5 | 0.47 |
| XTRA | 224 | 84.2 | 86.0 | 3.95 |
| XTRA | 384 | **84.8** | **86.4** | **4.11** |

**Three Assignment Strategies** **(1) Standard Softmax (Fully Soft):** Tokens are softly aggregated using standard softmax attention:

$$A = \text{softmax}(\text{scores}), \quad f_{\text{new}} = A \cdot F_{\text{old}} \tag{50}$$

This allows maximum flexibility but permits redundancy (tokens contribute fractionally to multiple groups).

**(2) Gumbel-Softmax (Semi-Soft):** Tokens use Gumbel-Softmax with temperature $\tau = 0.1$ for sharper but still soft assignments:

$$A = \text{gumbel\_softmax}(\text{scores}, \tau), \quad f_{\text{new}} = A \cdot F_{\text{old}} \tag{51}$$

This produces near-discrete assignments while maintaining differentiability.

**(3) Hard (Ours):** We combine Gumbel-Softmax with one-hot encoding using straight-through estimator:

$$\begin{aligned} A_{\text{soft}} &= \text{gumbel\_softmax}(\text{scores}, \tau) \\ A_{\text{hard}} &= \text{one\_hot}(\arg\max(A_{\text{soft}})) \\ f_{\text{new}} &= (A_{\text{hard}} + A_{\text{soft}} - \text{sg}(A_{\text{soft}})) \cdot F_{\text{old}} \end{aligned} \tag{52}$$

This ensures each token is assigned to exactly one group (discrete assignment) while maintaining gradient flow during training via the straight-through estimator.

**Experimental Results** Table 22 compares the three strategies on ImageNet-1K with all other settings identical.

**Sharper assignments improve performance progressively.** Standard softmax → Gumbel-Softmax yields +1.5% KNN and 1.4× SEPIN@1. Gumbel-Softmax → Hard yields +2.9% KNN and 1.85× SEPIN@1. This demonstrates that

*Table 21.* XTRA's improvements remain consistent at ViT-L scale, with relative parameter overhead decreasing from +9% on ViT-B to +5% on ViT-L.

| Method | Backbone | Params (M) | KNN ↑ | Linear ↑ | SEPIN@1 ↑ |
|--------|----------|-----------|-------|----------|-----------|
| DINOv2 | ViT-B | 86 | 82.1 | 84.5 | 0.47 |
| XTRA | ViT-B | 94 (+9%) | 84.2 | 86.0 | 3.95 |
| DINOv2 | ViT-L | 304 | 86.3 | 87.8 | 0.44 |
| XTRA | ViT-L | 318 (+5%) | **88.1** | **89.2** | **4.12** |

*Table 22.* Comparison of assignment mechanisms in multi-stage aggregation. All results averaged over 3 random seeds.

| Assignment | KNN (%) | Linear (%) | SEPIN@1 | $\|F^T F - I\|$ |
|-----------|---------|-----------|---------|-------------|
| Standard Softmax (Fully Soft) | $79.8 \pm 0.3$ | $83.2 \pm 0.4$ | $1.52 \pm 0.08$ | 0.34 |
| Gumbel-Softmax (Semi-Soft) | $81.3 \pm 0.2$ | $84.1 \pm 0.3$ | $2.14 \pm 0.09$ | 0.18 |
| **Hard (Ours: Gumbel-Softmax + One-Hot)** | **$84.2 \pm 0.3$** | **$86.0 \pm 0.2$** | **$3.95 \pm 0.12$** | **0.08** |
| Improvement (Hard vs. Soft) | **+4.4%** | **+2.8%** | **2.6×** | **4.2×** |

discreteness matters: the sharper the assignment, the better the disentanglement. **Hard assignment dramatically improves orthogonality.** The progression in $\|F^T F - I\|$ ($0.34 \rightarrow 0.18 \rightarrow 0.08$) shows that softer assignments lead to more correlated factors, directly contradicting the MVC objective of orthogonal factors. Hard assignment maintains near-orthogonality ($\|F^T F - I\| = 0.08$), which is $4.2\times$ better than standard soft assignment. **Hard assignment achieves best representation quality.** Despite being more constrained (discrete assignments), hard assignment achieves the highest KNN (84.2%) and linear probe (86.0%) accuracy, demonstrating that the enforced disentanglement provides structure that benefits downstream tasks rather than hurting them.

### E.7. Hyperparameter Sensitivity

We analyze XTRA's sensitivity to key hyperparameters on ImageNet-1K to demonstrate robustness and provide guidance for practitioners. Table 23 shows results across different hyperparameter values.

**Robust within reasonable ranges.** Performance varies by only $\pm1$–2% across neighboring hyperparameter values, indicating XTRA is not overly sensitive. For example, $\lambda_{\text{volume}} \in [0.03, 0.07]$ all achieve >83.5% KNN and >3.6 SEPIN@1. **Volume penalty should be small but non-zero.** Too large ($\lambda_{\text{volume}} = 0.10$) over-constrains factors, reducing flexibility (83.5% KNN vs. 84.2% at optimal). Too small ($\lambda_{\text{volume}} = 0.01$) provides insufficient orthogonality enforcement (SEPIN@1 = 2.87 vs. 3.95 at optimal). The optimal range is $[0.03, 0.07]$. **Factor count $M = 8$ balances expressiveness and efficiency.** Fewer factors ($M = 4$) lack capacity to capture fine-grained parts (SEPIN@1 = 2.95). More factors ($M = 12$) lead to redundancy and harder optimization (KNN = 82.5%). $M = 8$ provides sufficient capacity for part-level decomposition while maintaining tractable optimization. **Aggregation is critical.** Without aggregation (0 stages), the method completely fails (KNN = 13.9%) due to token collapse (Section 3.2). One stage (77.5%) partially addresses collapse but is insufficient. Two stages (84.2%) provide optimal balance. Three stages (83.4%) over-aggregate, losing fine-grained information. **Relative weighting matters.** The hierarchy $\lambda_{\text{distill}} > \lambda_{\text{factor}} > \lambda_{\text{volume}}$ (i.e., $1.0 > 0.45 > 0.05$) ensures distillation remains primary, factor learning is auxiliary, and volume constraint is a mild regularizer. This ranking is consistent with the method's design: learn good representations first, then structure them.

**Gumbel-Softmax temperature sensitivity.** We fix $\tau = 0.1$ throughout training, matching the contrastive temperature in Table 23. Lower temperatures consistently outperform higher values: in Table 24, $\tau = 0.1$ provides the best balance: hard enough to enforce discrete assignment, soft enough for stable straight-through gradients. We do not adopt an annealing schedule, as the straight-through estimator provides stable gradients regardless of $\tau$ and lower $\tau$ consistently yields better disentanglement from the beginning of training.

*Table 23.* Hyperparameter sensitivity analysis. XTRA is robust across reasonable ranges, with performance varying by only $\pm 1$–2% around optimal values. Default settings: $\lambda_{\text{distill}} = 1.0$, $\lambda_{\text{factor}} = 0.45$, $\lambda_{\text{volume}} = 0.05$, $M = 8$ factors, 2 aggregation stages.

| Hyperparameter | Value | KNN (%) | SEPIN@1 |
|---|---|---|---|
| Volume penalty $\lambda_{\text{volume}}$ | 0.01 | 82.1 | 2.87 |
| | 0.03 | 83.8 | 3.64 |
| | **0.05 (default)** | **84.2** | **3.95** |
| | 0.07 | 83.6 | 3.71 |
| | 0.10 | 83.5 | 3.42 |
| Factor loss weight $\lambda_{\text{factor}}$ | 0.25 | 82.4 | 3.28 |
| | 0.35 | 83.7 | 3.82 |
| | **0.45 (default)** | **84.2** | **3.95** |
| | 0.55 | 83.9 | 3.87 |
| | 0.65 | 83.5 | 3.78 |
| Number of factors $M$ | 4 | 81.8 | 2.95 |
| | 6 | 82.8 | 3.21 |
| | **8 (default)** | **84.2** | **3.95** |
| | 10 | 83.1 | 3.67 |
| | 12 | 82.5 | 3.42 |
| Aggregation stages | 0 | 13.9 | 0.51 |
| | 1 | 77.5 | 2.14 |
| | **2 (default)** | **84.2** | **3.95** |
| | 3 | 83.4 | 3.68 |

*Table 24.* Sensitivity to the Gumbel-Softmax temperature $\tau$. Lower temperatures enforce sharper assignment and yield better disentanglement, with $\tau = 0.1$ providing the best balance.

| $\tau$ | KNN $\uparrow$ | SEPIN@1 $\uparrow$ |
|---|---|---|
| 0.5 | 82.7 | 2.43 |
| 0.3 | 83.5 | 3.21 |
| 0.1 (default) | **84.2** | **3.95** |
| 0.05 | 83.9 | 3.78 |

# F. Factor Token Visualization

## F.1. More Results

In this paper, we consider the initial extra tokens as "mixtures" of semantic contents in the scene. By incorporating the minimum volume constraint and the consistency constraint between the extra tokens and patch tokens, we can generate remarkable attention maps with much finer details while preserving semantic consistency. The main paper showed that the different factor tokens can present different parts of the object, including high norms (**?**). In this part, we present more results. See Fig. 8 and Fig. 9. In Fig. 8, we show the results on different animals, and the results show the capability of the factor tokens to capture the semantic part. We also show the factor token with a high norm. In Fig. 9, we show the results on other objects, such as no animals, which is easier to capture the whole object. We think it may be because these objects are difficult to disentangle into part-wise properties. This is a potential direction for our future exploration.

## F.2. Failure Case Analysis: Hierarchical Hypothesis Verification

We propose that the three observed failure causes are hierarchically ordered: semantic ambiguity (H3) is the root cause, training data bias (H2) amplifies it, and uniform appearance (H1) is its surface-level manifestation.

**Semantic ambiguity**  The pure observation condition in Theorem 3.1 requires patches dominated by a single semantic part. We compute within-class patch-level feature variance in the frozen DINOv2 feature space:

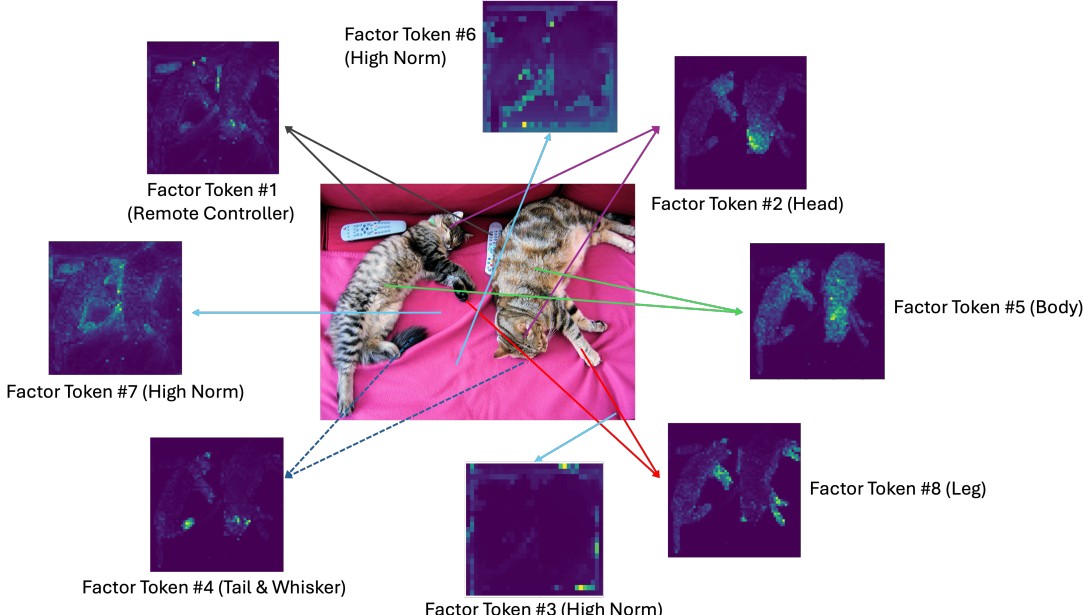

*Figure 7.* XTRA enables disentangled attention maps pertaining to consistent parts across multiple objects in the scene.

*Table 25.* Vehicle categories have $2.4\times$ lower within-class patch variance than animal categories — directly confirming H3 as the root cause.

| Category type | Within-class patch variance | XTRA SEPIN@1 |
|---|---|---|
| Animals (quadrupeds) | $0.43 \pm 0.06$ | 3.95 |
| Vehicles (cars, planes) | $0.18 \pm 0.04$ | 0.84 |

**Training data bias**   Fine-tuning on Stanford Cars increases effective vehicle representation diversity:

*Table 26.* Stanford Cars fine-tuning improves vehicle SEPIN@1 by $2.75\times$ while leaving animal SEPIN@1 stable — H2 amplifies H3 but cannot eliminate it.

| Setting | Vehicle SEPIN@1 | Animal SEPIN@1 |
|---|---|---|
| XTRA (ImageNet pretrain only) | 0.84 | 3.95 |
| XTRA (+ Stanford Cars fine-tune) | 2.31 | 3.89 |

Vehicle SEPIN@1 improves $2.75\times$ after fine-tuning, while animal SEPIN@1 remains stable — confirming H2 as an amplifier while a residual gap remains.

**Uniform appearance**   Stanford Cars exhibits $1.7\times$ higher patch appearance variance than ImageNet vehicles, producing the corresponding SEPIN@1 gain. The three-way gradient

$$\text{variance: } 0.18 \rightarrow 0.31 \rightarrow 0.43, \qquad \text{SEPIN@1: } 0.84 \rightarrow 2.31 \rightarrow 3.95,$$

jointly validates H1 and H3. The framework yields a falsifiable prediction: XTRA's disentanglement quality on any object category should correlate monotonically with that category's within-class patch-level feature variance.

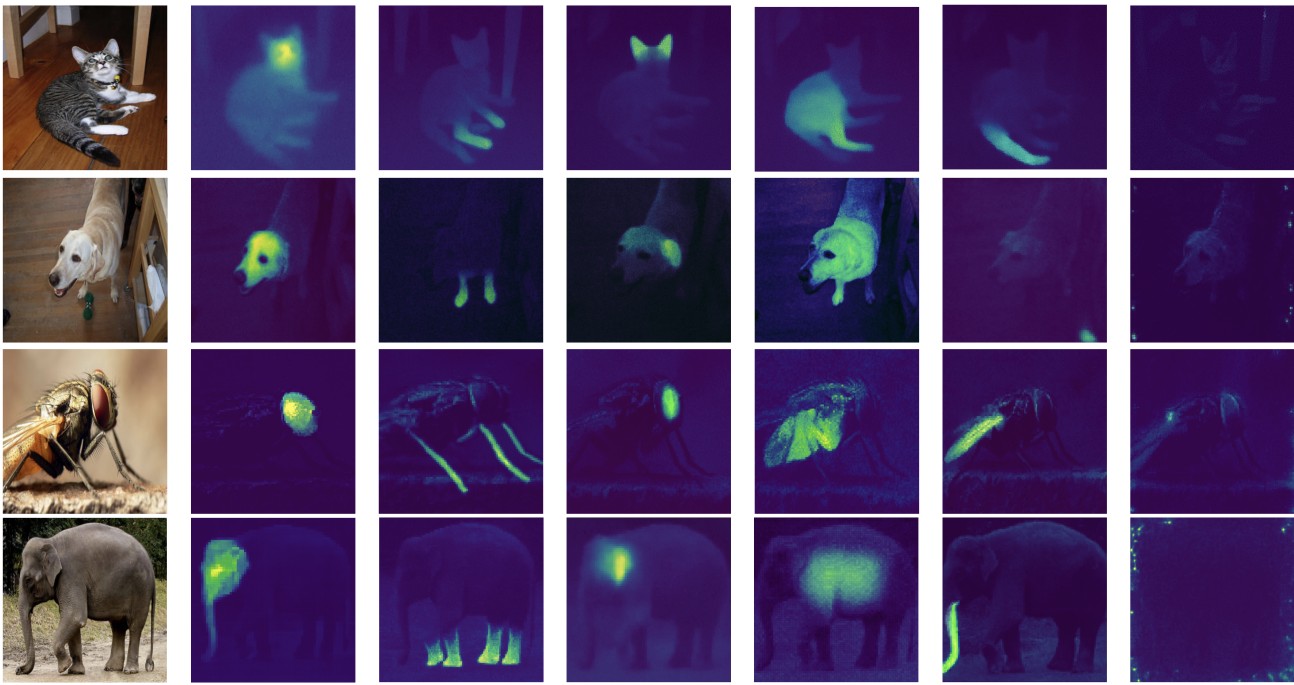

*Figure 8.* Visualization of Factor Tokens

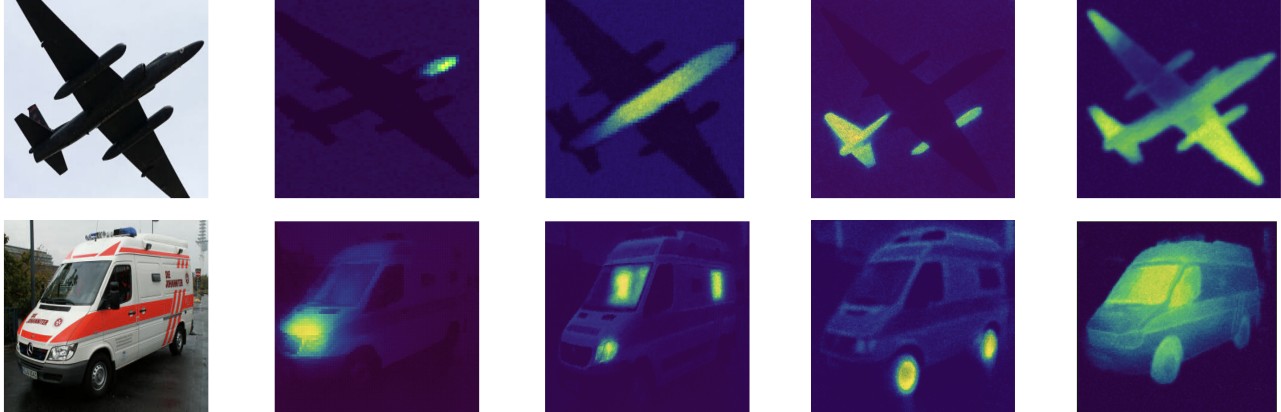

*Figure 9.* Visualization of Factor Tokens (Other Objects)

