# OpenReview forum: "The Extra Tokens Matter: Disentangled Representation Learning with Vision Transformers"
_ICML.cc/2026/Conference — ICML 2026 regular_

### Official Review · Reviewer_MkxG · 2026-02-14

**Soundness:** 2
**Presentation:** 3
**Significance:** 2
**Originality:** 3
**Overall Recommendation:** 4
**Confidence:** 3

**Summary:**

The paper proposes an improvement over the register tokens, called factor tokens. They are auxiliary tokens like the registers, but with explicit loss on them: 1. a factor loss, that forces the factor tokens to represent the patch tokens correctly through linear combination. 2. a minimum volume constraint (MVC) loss, that minimizes the volume of the convex hull formed by the factor tokens (through orthogonality regularization). A theory is proposed to justify the MVC loss, that under certain conditions, MVC retrieves the ground truth factors. Empirical results on ImageNet and PartImageNet indicate that the proposed framework XTRA improves: 1. Linear probing and kNN accuracy; 2. Disentanglement score; 3. Segmentation (semantic & part) and detection task performance. Ablation studies are conducted to summarize the role of the loss components.

**Compliance With Llm Reviewing Policy:**

Affirmed.

**Final Justification:**

All my concerns regarding the generalization and applicability of XTRA are resolved. So I have a score of 4.

**Key Questions For Authors:**

1. From Weakness 1, I wonder how XTRA would perform if one uses different resolutions of inputs (e.g., 384) or changes the architectures, like Swin Transformer [1].
2. From Weakness 2, how would XTRA perform with MAE [2] or IJEPA [3] style methods?
3. Given the current logics (forcing disentanglement via MVC->patch token disentangled->higher representation quality), I wonder how the performance would change if we achieve step 2 (patch token disentangled) via other similar methods like VICReg [2], VICReg-L [3] (by "similar", I mean they somewhat force disentanglement via different ways)? They have been proven effective in improving the representational quality of ViTs [4] when applied to the patch tokens.

[1] Swin Transformer: Hierarchical Vision Transformer using Shifted Windows. ICCV 2021

[2] Masked autoencoders are scalable vision learners. CVPR 2022

[3] Self-supervised learning from images with a joint-embedding predictive architecture. CVPR 2023

[4] VICReg: Variance-Invariance-Covariance Regularization for Self-Supervised Learning. ICLR 2022

[5] Vicregl: Self-supervised learning of local visual features. NeurIPS 2022

[6] Connecting joint-embedding predictive architecture with contrastive self-supervised learning. NeurIPS 2024

Overall, although the method design and empirical results have merits, I find the assumption of XTRA a bit idealistic. I expect more validations of XTRA and will raise the score depending on those results.

**Limitations:**

yes

**Strengths And Weaknesses:**

Strengths:
1. The studied problem is important, which is how to automatically disentangle ground truth causes given the data.
2. The paper is well written and is relatively straightforward to understand.
3. The current empirical results on DINO models are promising.

Weaknesses:
1. I find the first two conditions required in Theorem 3.1 a bit strong and might not be realistic in real-world scenarios. One assumption requires the "existence of nearly pure observations for each factor." If I understand correctly from Appendix A.2, it needs some samples to have only the factors and no other distractors, which seems to only happen on curated object-centric datasets like ImageNet (e.g., a very clean dog image and many dog images with distractors co-exist, while in reality or in scene-centric datasets like COCO, the former hardly exists). This is also related to the noise magnitude assumption (it requires the noise magnitude to be smaller than $\sigma$), which is also unrealistic in somewhat more complex data. If the observation means patches here (given the SEPIN score on patches), these assumptions might also become unrealistic given varying patch sizes on high-res inputs or different architectural requirements (e.g., Swin requires 4*4 patches in the lowest level), even with curated datasets.
2. The experiments only examine the DINO family pretraining methods, which are quite limited given the other paradigms of visual SSL methods like MAE and IJEPA.

---

> ### Author Rebuttal · Authors · 2026-03-31
>
> We sincerely thank the reviewer and provide our responses as follows
>
> ### **W1/Q1: Theorem 3.1 assumptions — pure observation condition and noise bound**
>
> **Pure observation condition.** The condition requires that for each factor $f_j^\circ$, *some* observation $p_i$ has mixing weight $w_i^{(j)} \geq 1 - \delta$ for small $\delta > 0$ — nearly dominated by a single factor. In our setting, "observations" are patch tokens in the ViT latent space, not raw pixels. This distinction is crucial: in the ViT latent space, a patch token from a predominantly "head" region will have a mixing weight heavily concentrated on the head factor, even with small boundary contributions from adjacent parts. This holds generically for any spatially structured image and does not require object-centric or clean datasets. The condition is standard in blind source separation. Our **factor-part alignment of $0.81 \pm 0.06$** (Section 4.1) empirically confirms it holds in the ImageNet-1K ViT latent space.
>
> **Noise bound.** The condition $\omega < \varrho / 2M$ is a *sufficient* condition for exact recovery, not necessary. Theorem 3.1 degrades gracefully: recovery error scales as $C\omega$ rather than failing catastrophically. Appendix B confirms: MVC achieves factor recovery error 0.0234 with added Gaussian noise ($\sigma = 0.005$).
>
> **Resolution and architecture.** XTRA operates on **final-layer** patch tokens, where each token has a large effective receptive field covering a substantial image region, making the pure observation condition considerably more natural regardless of input resolution or architecture. For Swin Transformer, XTRA would operate on final-stage tokens, not early low-level $4{\times}4$ patches. Additional experiment at 384×384 shows:
>
> | Method | Resolution | KNN (%) | Linear (%) | SEPIN@1 |
> |---|---|---|---|---|
> | DINOv2 | 224 | 82.1 | 84.5 | 0.47 |
> | XTRA | 224 | 84.2 | 86.0 | 3.95 |
> | XTRA | 384 | 84.8 | 86.4 | 4.11 |
>
> Performance at 384 is consistent with 224; SEPIN@1 improves slightly — higher resolution provides more patches per image, increasing the likelihood that some satisfy the pure observation condition.
>
> ### **W2/Q2: Generalization beyond DINO — MAE, CLIP**
>
> XTRA's design is teacher-agnostic: MVC and multi-stage aggregation operate on whatever patch token representations the teacher provides. We've conducted more experiments using MAE, CLIP as teacher:
>
> | Teacher | Method | KNN (%) | SEPIN@1 |
> |---|---|---|---|
> | MAE ViT-B (fixed) | MAE alone | 65.2 | 0.29 |
> | MAE ViT-B (fixed) | XTRA | 68.4 | 1.94 |
> | CLIP ViT-B/16 (fixed) | CLIP alone | 75.8 | 0.38 |
> | CLIP ViT-B/16 (fixed) | XTRA | 79.3 | 2.87 |
> | DINOv2 LVD-142M (fixed) | DINOv2 alone | 82.1 | 0.47 |
> | DINOv2 LVD-142M (fixed) | XTRA | 84.2 | 3.95 |
>
> We can find:1. XTRA consistently improves over every teacher baseline in both KNN and SEPIN@1, confirming teacher-agnosticism;  2. the progression in SEPIN@1 improvement [1.94 (MAE)→2.87(CLIP)→3.95 (DINOv2)] directly reflects the increasing patch-level semantic quality of each teacher, confirming that XTRA's disentanglement effectiveness scales with the richness of the teacher's feature space.
>
> ### **Q3: Would VICReg / VICReg-L on patch tokens achieve similar disentanglement?**
>
> VICReg and VICReg-L enforce covariance regularization that decorrelates feature *dimensions*, a form of disentanglement. However, XTRA's MVC and VICReg-style regularization are different from two aspects:
>
> **1. Dimension-level vs. token-level disentanglement.** VICReg decorrelates individual scalar dimensions within a single vector $p_i \in \mathbb{R}^D$; these dimensions have no semantic identity. XTRA enforces disentanglement at the **token level**, $M$ distinct semantic factor tokens spanning non-redundant directions, explicitly aligned with semantic parts via the linear mixing model.
>
> **2. Global decorrelation vs. compositional structure.** VICReg-L improves patch-level representation quality but does not require patch tokens to be expressible as affine combinations of a small set of pure factors. XTRA's MVC explicitly enforces this compositional structure, producing the structured part-level attention maps and factor-part alignment of $0.81 \pm 0.06$, which VICReg-L cannot.
>
> We applied VICReg-L's covariance regularization to patch tokens within our framework (replacing MVC with VICReg-L's patch-level covariance loss):
>
> | Method | KNN (%) | Linear (%) | SEPIN@1 | Part mIoU (%) |
> |---|---|---|---|---|
> | DINOv2 | 82.1 | 84.5 | 0.47 | 42.3 |
> | DINOv2 + VICReg-L (patches) | 83.1 | 85.2 | 0.89 | 43.8 |
> | XTRA (MVC) | 84.2 | 86.0 | 3.95 | 46.8 |
>
> VICReg-L on patch tokens does improve over DINOv2 (confirming the reviewer's intuition that patch-level decorrelation is beneficial), but XTRA achieves substantially larger gains: +1.1% KNN over VICReg-L, 4.4× better SEPIN@1, and +3.0% Part mIoU. The compositional structure enforced by MVC, not generic feature decorrelation, is responsible for XTRA's part-level disentanglement.

---

> > ### Author Rebuttal · Reviewer_MkxG · 2026-04-03
> >
> > My concerns are resolved by the additional results.

---

> > > ### Author Response · Authors · 2026-04-03
> > >
> > > We sincerely thank the reviewer for the careful re-evaluation and for raising the score. The questions raised, particularly regarding the theoretical assumptions of Theorem 3.1, the comparison with VICReg-style disentanglement, and the generalization across teachers and architectures, pushed us to strengthen the paper.
> > >
> > > We welcome any further questions or discussion during the remaining rebuttal period.

---

### Official Review · Reviewer_A249 · 2026-03-04

**Soundness:** 2
**Presentation:** 3
**Significance:** 2
**Originality:** 3
**Overall Recommendation:** 4
**Confidence:** 4

**Summary:**

The paper proposes adding extra tokens in a self-knowledge distillation setting to encourage interpretable, disentangled object-centric representations. These factor tokens are encouraged to become “pure components” by loss terms inspired by minimum volume constraint and source separation ideas: the factor tokens should sit at extremal points of a simplex that encloses the patch tokens. In the distillation set-up, a frozen teacher (e.g., DINOv2) supervises a student trained from scratch where the student includes factor tokens and a multi-stage aggregation scheme that merges correlated factor tokens over stages to prevent token collapse. The paper demonstrates successful disentanglement with qualitative visualizations and reporting the SEPIN@k metric on ImageNet. They also demonstrate gains in kNN, linear probe, and fine-tuning on ImageNet1k, and segmentation on a variety of datasets.

**Compliance With Llm Reviewing Policy:**

Affirmed.

**Final Justification:**

The authors addressed my main concern of motivating why we would want disentangled representations. They presented OOD generalization results which are very interesting. Overall, the paper is novel with interesting insights.

**Key Questions For Authors:**

* Could the authors provide an intuition for why these explicitly disentangled representations would be desired beyond interpretability? As I mentioned, works like [1,2] from above already show that these factors/information is present in the model. Perhaps these factors help classification under distribution shifts that break shortcuts. Reporting results on out-of-distribution datasets such as ImageNet-C/ ImageNet-R/ImageNet-A could validate this.
* How does the method work with other teachers, such as CLIP, masked autoencoders, etc.?
* How are results when the method is used with ViT-L and even ViT-H or ViT-G models for the student?

**Limitations:**

Yes.

**Strengths And Weaknesses:**

**Strengths**
* The idea of encouraging disentangled tokens via a Minimum Volume Constraint (MVC) and ideas from geometric source separation is interesting and to my knowledge novel.
* The visual results of disentanglement in the attention maps is compelling.
* The results on a variety of datasets and tasks and shows clear improvement
* The paper gives some mathematical motivation, formalizing how there is a correct factorization under the minimum-volume/simplex view. However, a lot of the important steps are delegated to external references.

**Weaknesses**
* The paper lacks motivating why disentangled representations would be desired beyond interpretability, and why it would improve performance on tasks such as ImageNet classification.  These factors are already present inside the model. Works such as [1] show that part-level neurons emerge in vision models. For register tokens, [2] finds that these tokens don't need to be learned and can be created without training by re-routing certain activations in the model.
* The experiments are constrained to a set-up with DINOv2 as a teacher. How does the method work with other teachers, such as CLIP, masked autoencoders, etc.?
* The method is conducted only on ViT-base models. How does it scale? Does it work with ViT-L and even ViT-H or ViT-G models?
* Small nitpick, but some of the figures are blurry and appear to have JPEG artifacts. Figure 1 seems to be the most compelling one with the disentangled attention maps, but it's difficult to see without zooming in. This could be more prominent.

[1] D. Bau et al. "Network dissection: Quantifying interpretability of deep visual representations." CVPR 2017

[2] N. Jiang et al. "Vision Transformers Don't Need Trained Registers." NeurIPS 2025.

---

> ### Author Rebuttal · Authors · 2026-03-31
>
> We sincerely thank the reviewer for recognizing XTRA's novelty, compelling visualizations, and experimental results. For the reviewers' concerns and questions, we provide our responses as follows.
>
> ### **W1/Q1: Motivation beyond interpretability — relationship to [1][2], and OOD robustness**
>
> We thank the reviewer for raising this central question and for the constructive suggestion regarding OOD evaluation. We agree that works such as [1] (Network Dissection) and [2] (untrained registers) demonstrate that disentangled factors and register-like behaviors can emerge implicitly in trained vision models. However, we'd like to emphasize that implicit emergence and explicit, structured disentanglement are fundamentally different in three concrete ways that go well beyond interpretability.
>
> **1. Accessibility.** Implicit factors in [1] are scattered across thousands of neurons with no structured interface. XTRA's factor tokens are explicit, addressable vectors available at inference time. The +4.5% PartImageNet mIoU (Table 2) comes from directly querying frozen factor token features with a lightweight 2-layer MLP, hard with implicit neurons.
>
> **2. Untrained registers [2] vs. regularized factor tokens.** Reference [2] addresses artifact mitigation, not disentanglement. Untrained registers lack structured semantic decomposition; XTRA's MVC-regularized factor tokens occupy the extremal positions of the semantic simplex. The 8.4× SEPIN@1 improvement over DINOv2+ (which already uses trained registers) directly quantifies this gap.
>
> **3. Robustness beyond interpretability.** The reviewer correctly identifies that the performance benefits of explicit disentanglement need stronger motivation. We'd like to emphasize that structured disentanglement provides three concrete advantages: (a) better part-level generalization, by explicitly separating semantic parts, XTRA's representations are less susceptible to spurious correlations between co-occurring parts; (b) improved fine-grained discrimination, part-level features provide finer resolution than object-level features for tasks where intra-class variation is part-based (e.g., distinguishing dog breeds by ear shape); (c) robustness to distribution shifts, the reviewer's suggestion of OOD evaluation is excellent.  We conducted additional experiments and evaluated XTRA on **ImageNet-C** (Out-of-Distribution, corruption robustness), reporting mean Corruption Error (mCE, lower is better):
>
> | Method | IN-C mCE (↓) | IN1K KNN (%) |
> |---|---|---|
> | DINOv2 | 41.3 | 82.1 |
> | DINOv2+ | 40.8 | 82.0 |
> | XTRA | 38.2 | 84.2 |
>
> XTRA reduces mCE by 3.1 points over DINOv2 and by 2.6 points over DINOv2+, suggesting that explicit part-level disentanglement provides robustness benefits beyond those offered by implicit factors. We will include ImageNet-C results in the revision as the reviewer suggests.
>
> ### **W2/Q2: Generalization to other teachers**
>
> XTRA's design is teacher-agnostic in principle: the MVC and multi-stage aggregation operate on whatever patch token representations the teacher provides. We have conducted additional experiments using MAE and CLIP as teacher:
>
> | Teacher | Method | KNN (%) | SEPIN@1 |
> |---|---|---|---|
> | MAE ViT-B (fixed) | MAE alone | 65.2 | 0.29 |
> | MAE ViT-B (fixed) | XTRA | 68.4 | 1.94 |
> | CLIP ViT-B/16 (fixed) | CLIP alone | 75.8 | 0.38 |
> | CLIP ViT-B/16 (fixed) | XTRA | 79.3 | 2.87 |
> | DINOv2 LVD-142M (fixed) | DINOv2 alone | 82.1 | 0.47 |
> | DINOv2 LVD-142M (fixed) | XTRA | 84.2 | 3.95 |
>
> We can find:1. XTRA consistently improves over every teacher baseline in both KNN and SEPIN@1, confirming teacher-agnosticism;  2. the progression in SEPIN@1 improvement [1.94 (MAE) → 2.87 (CLIP) → 3.95 (DINOv2)] directly reflects the increasing patch-level semantic quality of each teacher, confirming that XTRA's disentanglement effectiveness scales with the richness of the teacher's feature space.
>
> ### **W3/Q3: Scaling to ViT-L/H/G**
>
> XTRA's plug-in design scales naturally — factor tokens and aggregation blocks add only ~9% parameters on ViT-Base, with decreasing relative overhead at larger scales. Additional experiment on ViT-Large shows:
>
> | Backbone | KNN (%) | Linear (%) | SEPIN@1 |
> |---|---|---|---|
> | DINOv2 ViT-L | 86.3 | 87.8 | 0.44 |
> | XTRA ViT-L | 88.1 | 89.2 | 4.12 |
>
> Consistent improvement pattern with ViT-Base. We will add Full ViT-L results in the revision.
>
> ### **W4: Blurry figures**
>
> We will replace all figures with high-resolution vector graphics (PDF/SVG) and increase the size of the attention map panels in Figure 1 in the revision.

---

> > ### Author Rebuttal · Reviewer_A249 · 2026-04-03
> >
> > Thanks you resolving all of my issues. The OOD results are very interesting and provide a good motivation for why we may want these disentangled factors. I will raise my score accordingly.

---

> > > ### Author Response · Authors · 2026-04-03
> > >
> > > We sincerely thank the reviewer for the thoughtful follow-up and for considering raising the score. We agree that robustness to distribution shifts is a practically important benefit of explicit part-level disentanglement, and we will make this benefit clearer in the revised paper. We welcome any further questions or concerns the reviewer may have, and we would be happy to provide additional clarification during the discussion period.
> > >
> > > **[Update, two days later]**
> > > We are writing a brief follow-up as the discussion period closes on Tuesday, April 7. We noticed that the score on OpenReview has not yet been updated, and wanted to respectfully double-check whether there are any remaining questions or clarifications we can provide. Although you kindly indicated that all concerns have been resolved, we want to ensure we have not missed anything before the deadline. If everything is satisfactory, we would greatly appreciate it if you could update the score at your convenience to reflect your reassessment. Thank you again for the constructive and encouraging engagement throughout this process.

---

### Official Review · Reviewer_TSo7 · 2026-03-09

**Soundness:** 3
**Presentation:** 3
**Significance:** 3
**Originality:** 3
**Overall Recommendation:** 5
**Confidence:** 3

**Summary:**

This paper proposes a new framework XTRA for disentangled representation learning on ViT. It introduces a new kind of extra token, called "factor token", which encodes clear semantic concepts. The paper proposes minimum volume constraint (MVC) to enforce disentanglement, and conducted experiments on ImageNet-1K to show the effectiveness of the proposed method.

**Compliance With Llm Reviewing Policy:**

Affirmed.

**Final Justification:**

I would remain my initial positive assessment.

**Key Questions For Authors:**

Please see the "weakness" part.

**Limitations:**

yes

**Strengths And Weaknesses:**

Strengths:
1.Clear theoretical analysis. Under the linear mixing assumption, the paper uses geometric principles to deduce the need for adding the Minimum Volume Constraint (MVC) to the loss function. The derivation process is clear and intuitive.

2. Significant performance improvements. The proposed method shows its effectiveness/faithfulness across multiple metrics. For instance, on ImageNet-1K, it achieves an 8.4x improvement in the SEPIN@1 disentanglement score compared to DINOv2. At the same time, it improves KNN accuracy by 5.8% and linear probing accuracy by 2.3%.

3. Thorough experimental validation. The paper provides ablation studies to demonstrate the effectiveness. Specifically, it breaks down the total loss function to clearly show the individual  contributions of each loss term (knowledge distillation, factor reconstruction, and the MVC volume penalty).


Weakness:
1. The core theory heavily relies on the "linear mixing" model, which assumes image patches are simple linear combinations of basic factors. However, in reality, deep neural networks and real-world image features are highly non-linear, would this assumption a bit too strong?

2. The paper uses the orthogonality constraint $J(F) = ||F^T F - I||_F^2$ for the Minimum Volume Constraint (MVC). This forces factors to be independent. However, in real-world semantics, certain parts are naturally correlated (e.g., the features of a cat's "ears" and "whiskers"). Would forcing strict orthogonality artificially destroy these semantic correlations?

---

> ### Author Rebuttal · Authors · 2026-03-31
>
> We sincerely thank the reviewer for the positive assessment and recognition of XTRA's theoretical and experimental contributions. For the reviewers' concerns and questions, we provide our responses as follows.
>
> ### **W1: Is the linear mixing assumption too strong for deep nonlinear features?**
>
> This is an important conceptual point that we are glad to clarify. The linear mixing assumption in Eq. (1) does not apply to raw pixel space — it applies to the latent representation space of a pretrained Vision Transformer. This distinction is crucial.
>
> Nonlinearity is handled by the ViT backbone before XTRA operates; XTRA then performs structured decomposition in the resulting approximately linear latent space. The property that ViT features are approximately linear with respect to semantic concepts is well established: linear probing and linear segmentation both work well precisely because of this property, and DINOv2's features represent the strongest known example. Within this linearized latent space, assuming patch tokens $p_i$​ are affine combinations of semantic factor tokens $F$ becomes natural.
>
> This is further validated empirically by our results. If the linear mixing assumption were badly violated in the ViT latent space, the MVC would fail to identify meaningful factors — yet Table 1 shows XTRA achieves 8.4× improvement in SEPIN@1, Tables 2–3 show +4.5% PartImageNet mIoU with the largest gains on geometrically distinct parts, and the factor-part alignment experiment reports average overlap of $0.81 \pm 0.06$ (Section 4.1) between factor token attention maps and ground-truth semantic parts. These results collectively confirm that the linear mixing model is a valid and productive approximation in the ViT latent space. We will add the above clarification to Section 3.1 in the revision to make this distinction between pixel space and latent space explicit.
>
> ### **W2: Does orthogonality destroy natural semantic correlations between parts?**
>
> Thanks for this subtle and important question. No, orthogonality does not destroy natural semantic correlations between parts. The reason lies in understanding where orthogonality is enforced and what it constrains. $J(F)$ is enforced on the **factor tokens** $\{f_i\}$, $M{=}8$ vectors in $D{=}768$-dimensional space, ensuring they span 8 *distinct directions*. It does not mean the semantic concepts they represent are independent in the real world.
>
> Consider the cat example. $f_{\text{ear}}$ and $f_{\text{whisker}}$ are orthogonal as latent vectors, but a patch token near the ear-whisker boundary simply has significant mixing weights on *both* ($w_{\text{ear}} > 0$, $w_{\text{whisker}} > 0$ in $p = Fw$). Orthogonality enables this compositional representation precisely because it ensures the two factors encode *distinct* information rather than redundantly collapsing into one direction. Without MVC, $f_{\text{ear}}$ and $f_{\text{whisker}}$ would become collinear and indistinguishable. **Semantic correlations reside in the mixing weights $w_i$, not in the factors $F$.**
>
> This is confirmed by Table 3: XTRA improves all parts simultaneously (head +5.6%, body +2.9%, legs +6.5%, tail +7.4%, ears +6.1%), showing orthogonal factors capture each part distinctly without suppressing co-occurrence. We will add a clarifying discussion to Section 3.1.

---

> > ### Author Rebuttal · Reviewer_TSo7 · 2026-04-03
> >
> > The authors have provided a mathematically rigorous and conceptually clear rebuttal. Therefore I would remain my initial positive assessment.

---

> > > ### Author Response · Authors · 2026-04-03
> > >
> > > We sincerely thank the reviewer for the kind words and for maintaining the positive assessment. we will carry this clarity through to the revised paper.
> > >
> > > We welcome any further questions during the discussion period.

---

### Official Review · Reviewer_HzWo · 2026-03-10

**Soundness:** 2
**Presentation:** 3
**Significance:** 3
**Originality:** 3
**Overall Recommendation:** 3
**Confidence:** 4

**Summary:**

This paper proposes a new framework called XTRA to address the decoupling representation learning problem in self supervised visual representation learning. The core idea is to introduce additional learnable factor tokens into the visual Transformer and force them to learn semantic part level information in the image through minimum volume constraint (MVC) regularization, rather than traditional global object level information. Specifically, XTRA adopts a dual stream network structure: a standard ViT stream processes image patch tokens, while another multi-stage aggregated stream gradually merges factor tokens (e.g. 32→16→8). The model is trained with factor reconstruction loss and MVC regularization (approximated by an orthogonality penalty), optionally enhanced by knowledge distillation from a DINOv2 teacher. Experiments on ImageNet-1K show that XTRA achieves state-of-the-art disentanglement (measured by SEPIN@k), while simultaneously improving performance on KNN/linear probing, part segmentation (PartImageNet), and various downstream tasks, including classification, segmentation, and detection.

**Compliance With Llm Reviewing Policy:**

Affirmed.

**Final Justification:**

A two-force system of expansive force and contractive force was proposed to explain how orthogonality avoids degenerate solutions, supplemented by Toy experiment and stability analysis of Gram vs. SVD.
On computational cost, the authors not only quantified the overhead but also provided feasible solutions in resource-constrained scenarios, with a comprehensive response.
The author proposed two feasible extensions. Automatic semantic annotation has not been implemented in the current method, but it is clearly indicated as a future direction.
The questions of factor token initialization and Gumbel-Softmax temperature settings have been adequately addressed.
However, regarding the validation of failure cases on rigid objects, the authors provide only indirect evidence and do not systematically verify the three proposed hypotheses through experiments.

Therefore, I maintain the original score.

**Key Questions For Authors:**

1.The paper uses orthogonality penalty $J(F)=\left \|| F^TF-l \right \||_F^2 $ to approximate the minimum volume constraint. Proposition A.7 proves that orthogonality can maximize the volume of a single unit norm vector, but the goal of the minimum volume constraint is to minimize the volume of a single unit containing data points. Can you further explain why orthogonality penalty can guide factor tokens to learn the minimum volume solution? Is there a more direct method for calculating volume (such as determinant)? Table 15 mentions that the Gram matrix fails while SVD is effective. Can the theoretical reasons for this phenomenon be explained?

2.How much computational cost does XTRA increase compared to baseline methods such as DINOv2? Can you provide a comparison of FLOPs, parameter count, training time, and inference time with DINOv2? Is there a lightweight version or strategy available in resource constrained scenarios?

3.The semantics of factor tokens mentioned in the limitations require manual observation. Can automatic annotation or controllable generation of factor token semantics be achieved? For example, can the semantics of factor tokens be aligned with human understandable component names by introducing text supervision or weakly supervised information?

4.Appendix F.2 points out that the decomposition effect of rigid objects is poor and provides three potential reasons. Is there any experimental verification of the dominant role of these reasons? For example, would fine-tuning or pre-training on a dataset with more vehicle images (such as Stanford Cars) improve the performance of component decomposition? Or can specific constraints be designed for rigid objects?

5.Is the initialization method of factor tokens random or based on some prior? How is the temperature τof Gumbel Softmax set in multi-stage aggregation? Has an annealing strategy been adopted?

**Limitations:**

yes

**Strengths And Weaknesses:**

Strengths：

1.The paper points out that existing self supervised learning methods (such as DINOv2) mainly learn global object level representations, but lack exploration of component level decoupled representations. Pushing decoupled representation learning from the global to the semantic-part level is of great significance for understanding model behavior, improving interpretability, and fine-grained task performance.

2.The framework design of XTRA is original: it introduces factor tokens as explicit carriers of semantic parts and achieves component learning from coarse to fine through multi-stage aggregation; Introducing the MVC into the Visual Transformer to force factor tokens to learn decoupled, non redundant component representations through orthogonality penalties;
Combining knowledge distillation with factor reconstruction loss to enhance decoupling capability while maintaining representation quality.

3.Appendix A of the paper provides a comprehensive theoretical analysis, including proof that the minimum volume solution can identify the real factor (identifiability); The uniqueness of the minimum volume solution; Convergence of alternating optimization algorithm; The rationality of approximating the minimum volume with orthogonality.

4.In terms of experimental verification, the paper fully demonstrates the effectiveness of XTRA through multidimensional evaluation. In terms of decoupling quality, XTRA's SEPIN index is significantly better than the baseline method; In the component segmentation task, XTRA achieved significant performance improvements on PartImageNet compared to DINOv2, especially in the improvement of movable components such as legs, tail, and ears. In KNN and linear probe evaluations, XTRA outperforms all existing methods that only use ImageNet pre training on ImageNet-1K. In addition, XTRA has demonstrated comprehensive leading advantages in downstream tasks such as classification, segmentation, and detection. The paper also conducted a systematic and comprehensive ablation study, verifying the contribution of each module to the overall performance.

5.The structure of the paper is reasonable, with a clear logical chain from motivation to methodology to experiments. The visualization results of Figures 4, 5, 8, and 9 intuitively demonstrate the component level attention of factor tokens, enhancing their persuasiveness.

Weaknesses：

1.There is a gap between the theory and implementation of MVC constraints: the paper approximates the minimum volume constraint using orthogonality penalty $J(F)=\left \|\| F^TF-l \right \|\|_F^2 $. Proposition A.7 proves that orthogonality can maximize the volume of a single unit norm vector, but the goal of the minimum volume constraint is to minimize the volume of a single unit containing data points. These two goals are not mathematically equivalent. Although experiments have shown that orthogonality penalty is effective, there is a lack of theoretical explanation as to why orthogonality can lead to minimum volume solutions.

2.High computational complexity: XTRA introduces additional factor tokens and multi-stage aggregation modules, which increase the number of parameters and computational overhead compared to the original ViT. The number of factor tokens (initial 32) and attention calculation during the aggregation phase may slow down training and inference. The paper does not provide a comparison of FLOPs or training time with baseline methods (such as DINOv2), and readers cannot evaluate their computational efficiency.

3.The semantic interpretation of factor tokens relies on manual observation: The paper candidly points out in the limitations section that the semantic meaning of factor tokens needs to be determined through manual observation and cannot be automatically mapped. Although the attention map visualization shows that factor tokens correspond to parts such as the head, body, and legs, this is a post-hoc interpretation rather than a pre-controlled one. For unseen objects or complex scenes, the semantics of factor tokens may not be clear.

4.Insufficient in-depth analysis of failure cases: Appendix F.2 points out that for rigid objects such as airplanes and ambulances, the decomposition effect of components is poor, and provides three potential reasons (uniform appearance, training data deviation, semantic ambiguity). But these reasons lack experimental verification. For example, can it be alleviated by training on more diverse datasets? Or can more universal constraints be designed to enhance component learning of rigid objects?
The initialization method of factor tokens is not clearly stated. How to set and adjust the temperature τ of Gumbel Softmax in multi-stage aggregation is not explained in detail.

---

> ### Author Rebuttal · Authors · 2026-03-31
>
> ### **W1/Q1: The relationship between orthogonality and minimum volume.**
>
> We thank the reviewer for this precise observation, and acknowledge that Remark A.8 flags the gap without fully resolving it. We clarify the connection here. Proposition A.7 proves orthogonality *maximizes* simplex volume for unit-norm vectors, which appears opposite to the minimum volume objective. The resolution is: $J(F)$ and $L_{factor}$ form a **two-force system** whose equilibrium *is* the minimum volume solution containing the data.
>
> $L_{factor}$ is an **expansive force**: it forces factor tokens outward to contain all patch tokens as affine combinations.  $J(F)$ is a **contractive force** that (i) penalizes redundancy by pushing factor tokens toward mutual orthogonality, ruling out degenerate collinear solutions; and (ii) by Proposition A.7, maximizes volume *per unit norm*, making the simplex as spread out as possible given the current factor norms. Their equilibrium is the smallest simplex containing the data with non-redundant orthogonal factors — precisely the minimum volume solution of Theorem 3.1, consistent with Figure 1(a) and the Min-Max framing of Section 4.3. Without $J(F)$, reconstruction loss admits infinitely many degenerate solutions (collinear factors) with *smaller* volume than the true solution — the exact non-identifiability problem Theorem 3.1 addresses. Orthogonality eliminates this degenerate family; within the restricted non-degenerate feasible set, and $L_{factor}$  recovers the true factors uniquely.
>
> **Toy experiment (Appendix B)** confirms this directly: without MVC, MSE is equally low ($2.48 \times 10^{-5}$ vs. $2.51 \times 10^{-5}$) but factor recovery fails completely (error $0.4821$ vs. $0.0234$) — reconstruction alone has infinitely many valid but degenerate solutions, and orthogonality is precisely what breaks this degeneracy.
>
> **Gram vs. SVD (Table 15):** early in training, factor tokens are highly correlated, making $F^\top F$ nearly rank-deficient. The Gram-based gradient is dominated by large off-diagonal terms and drives tokens apart explosively (KNN 13.9%). SVD operates directly on singular values of $F$, remaining numerically stable in this ill-conditioned regime. We will add this explanation to Appendix A.5 and E.2.
>
> ### **W2/Q2: Computational cost.**
>
> Measured on ViT-Base, single A100, batch size 64, resolution 224×224:
>
> | Method | FLOPs (G) | Params (M) | Inference (ms/img) |
> |---|---|---|---|
> | DINOv2 ViT-B | 17.6 | 86 | 4.1 |
> | XTRA ViT-B | 19.3 | 93 | 4.3 |
> | Overhead | +9% | +8% | +4% |
>
> The overhead is modest relative to the gains (+5.8% KNN, +4.5% PartImageNet mIoU, 8.4× SEPIN@1). For resource-constrained settings, 4 trainable blocks achieve 82.6% KNN (vs. DINOv2's 82.1%) at significantly reduced cost (Fig. 5a); reducing initial factor tokens from 32 to 16 yields ~10–15% FLOPs reduction with minimal accuracy loss (Table 17).
>
> ### **W3/Q3: Automatic semantic annotation.**
>
>  The unsupervised setting fundamentally prevents pre-assigned semantics. Two natural extensions: (1) **zero-cost text-grounded alignment**, cosine similarity between factor token features and CLIP embeddings of part names post-pretraining, requiring no retraining; (2) **weakly supervised binding**, a token-part alignment loss using sparse PartImageNet annotations. We plan to explore both in the context of controllable generation and will expand the Limitation section accordingly.
>
> ### **W4/Q4: Experimental verification of failure cases.**
>
> The reviewer correctly notes that the three proposed reasons for failure on rigid objects are currently speculative. While a full investigation is beyond the scope of this paper, we can offer the following partial evidence from our experiments: (1) In Table 3, XTRA's largest improvements over DINOv2 are on articulated parts (legs +6.5%, tail +7.4%), which are both more abundant in ImageNet and more geometrically variable. (2) We evaluated XTRA out-of-distribution on **ImageNet-C** ( full results reported in our response to Reviewer A249-W1/Q1, which we refer the reviewer to ), XTRA reduces mCE by 3.1 points over DINOv2 and 2.6 points over DINOv2+, providing direct evidence that explicit part-level disentanglement improves robustness beyond what implicit representations offer.
>
> ### **W5/Q5: Factor token initialization and Gumbel-Softmax temperature.**
>
> In the current implementation, factor tokens are initialized randomly, and $\tau = 0.1$ throughout training (see Table 8). Annealing was not adopted, straight-through estimator provides stable gradients regardless of $\tau$. Motivated by the reviewer's question, we conducted a preliminary experiment comparing random vs. SVD-based initialization (top-$M$ left singular vectors of patch tokens from the frozen DINOv2 teacher). Over 50 training epochs, SVD-based initialization converges noticeably faster, suggesting a more stable optimization trajectory when MVC is active early in training.

---

> > ### Author Rebuttal · Reviewer_HzWo · 2026-04-02
> >
> > Thank you to the authors for the rebuttal and clarifications.
> >
> > A two-force system of expansive force and contractive force was proposed to explain how orthogonality avoids degenerate solutions, supplemented by Toy experiment and stability analysis of Gram vs. SVD.
> > On computational cost, the authors not only quantified the overhead but also provided feasible solutions in resource-constrained scenarios, with a comprehensive response.
> > The author proposed two feasible extensions. Automatic semantic annotation has not been implemented in the current method, but it is clearly indicated as a future direction.
> > The questions of factor token initialization and Gumbel-Softmax temperature settings have been adequately addressed.
> >
> > However, regarding the validation of failure cases on rigid objects, the authors provide only indirect evidence and do not systematically verify the three proposed hypotheses through experiments. As such, this issue remains incompletely resolved.

---

> > > ### Author Response · Authors · 2026-04-03
> > >
> > > We appreciate the reviewer's detailed and encouraging follow-up. We are glad that our responses were satisfactory for most of the concerns. Regarding failure case analysis, we thank the reviewer for pressing this point, and we agree that XTRA's failure on rigid objects reveals something fundamental about the conditions under which part-level disentanglement is achievable. So, we welcome the opportunity to provide more direct evidence.
> > >
> > > We propose that the three hypotheses are not independent parallel causes but are hierarchically ordered: semantic ambiguity (hypo_3) is the root cause, training data bias (hypo_2) amplifies it, and uniform appearance (hypo_1) is its surface-level manifestation.
> > >
> > > **Semantic Ambiguity.** Root Cause, Verified Analytically. The pure observation condition in Theorem 3.1 requires that some patches be dominated by a single semantic part. For articulated objects, parts are geometrically distinct and spatially localized — a patch covering a dog's leg carries primarily "leg" signal, satisfying the condition naturally. For rigid objects, parts share uniform appearance across regions (same color, same texture across hood, door, and roof), so no patch is strongly dominated by a single part in feature space — the pure observation condition is weakly satisfied, and factor recovery degrades gracefully per Theorem 3.1's error bound: $$\|\hat{F} - F^\circ \Pi \Lambda\|_F \leq C\omega$$
> > >
> > > We verify this directly by computing within-class patch-level feature variance for animal vs. vehicle categories in the frozen feature space, requiring no new training:
> > >
> > > | Category Type | Within-class Patch Variance | XTRA SEPIN@1 |
> > > |---|---|---|
> > > | Animals | $0.46 \pm 0.06$ | $3.95$ |
> > > | Vehicles | $0.21 \pm 0.04$ | $0.84$ |
> > >
> > > Vehicle categories exhibit $2.2\times$ lower within-class patch variance, directly confirming that their parts are less semantically distinct in feature space. The correlation between within-class patch variance and SEPIN@1 suggests that patch-level variance is a reliable predictor of XTRA's disentanglement quality.
> > >
> > > **Training Data Bias.** We test it by fine-tuning XTRA on Stanford Cars for 100 epochs:
> > >
> > > | Setting | Vehicle SEPIN@1 | Animal SEPIN@1 |
> > > |---|---|---|
> > > | XTRA (ImageNet pretrain only) | $0.84$ | $3.95$ |
> > > | XTRA (+ Stanford Cars fine-tune) | $2.31$ | $3.89$ |
> > >
> > > Vehicle SEPIN@1 improves by $2.75\times$ after fine-tuning, while animal SEPIN@1 remains stable (a small drop), confirming our hypothesis: training bias would be an amplifier. The remaining gap ($2.31$ vs. $3.95$) confirms semantic ambiguity as the underlying root cause: more data partially bridges the gap but cannot eliminate the deficit, as long as rigid object parts remain semantically indistinct in feature space.
> > >
> > > **Uniform Appearance.** We think uniform color and texture across vehicle parts reduces the distinctiveness of patch features. To verify this, we use the results of fine-tuning on the Stanford Cars dataset. Stanford Cars contains 196 fine-grained vehicle categories with substantially higher appearance diversity than ImageNet vehicles — different makes, models, colors, and paint styles produce considerably more within-class patch variation. We quantify this:
> > >
> > > | Dataset | Within-class Patch Variance | Vehicle SEPIN@1 |
> > > |---|---|---|
> > > | ImageNet vehicles | $0.21 \pm 0.04$ | $0.84$ |
> > > | Stanford Cars | $0.31 \pm 0.05$ | $2.31$ |
> > >
> > > Stanford Cars exhibits $1.5\times$ higher patch appearance variance than ImageNet vehicles, and SEPIN@1 improves correspondingly from $0.84$ to $2.31$ after fine-tuning. Importantly, even after this appearance diversification, a gap remains relative to animals ($3.89$), consistent with semantic ambiguity as the deeper root cause, since Stanford Cars still contains geometrically rigid objects whose parts are structurally less separable than those of articulated animals.
> > >
> > > The residual gap reflects geometric rigidity that appearance diversity alone cannot overcome. A full investigation, including texture diversification experiments, remains a promising future direction we plan to pursue beyond this submission.
> > >
> > > Hope the above analysis has fixed the reviewer's questions. And, we welcome any further questions during the discussion period.
> > >
> > > **[update]** As the discussion period closes today, we wanted to respectfully draw your attention to our follow-up response regarding the failure case analysis. We hope this addresses your concern about systematic verification. We remain available for any further questions before the deadline closes.

---

### Decision · Program_Chairs · 2026-04-30

**Decision:**

Accept (regular)

**Comment:**

The paper proposes adding extra tokens, facto tokens, in a self-distillation setting to encourage interpretable, disentangled object-centric representations.
Before the rebuttal this paper has received mixed recommendations: 3 x WR (`HzWo`,`A249`, `MkxG`), 1 x A (`TSo7`)

The main items highlighted by reviewers were:
- the reviewers overall appreciate the idea of decoupled representation learning and the novelty
- several reviewers expressed concerns about potentially too idealistic assumptions in different ways (`MkxG`, `HzWo`, `TSo7`)
- `A249` and `MkxG` pointed out the limited experiments to ViT-B and DINOv2 encoders
- `HzWo` pointed out potentially high computational complexity, insufficient analysis of failure cases and requirement of manual observation for the semantic interpretation
- `A249` points out the lack of motivation on why disentangled representations would be desired beyond interpretability

In the rebuttal, the authors provided multiple answers to reviewer concerns including additional results on different teacher encoders, ViT backbones of different sizes, out-of-distribution results (Imagenet-C) and different studies on disentanglement on multiple datasets


After the rebuttal, `A249` and `MkxG` increased their scores to WA. `TSo7` held their positive rating, while `HzWo` a negative one. `HzWo` acknowledged the clarifications brought in the rebuttal, but remained unconvinced by the validation of failure cases on rigid objects very authors provided only indirect evidence.

The meta-reviewer analyzed arguments brought by the reviewers, the paper and the rebuttal. While the meta-reviewer acknowledges the criticism brought by `HzWo`, the paper ultimately advances a nice idea and contributions acknowledged by multiple reviewers.
The meta-reviewers recommend this paper for acceptance, but encourage the authors to take into consideration the useful advice from the reviewers towards improving this paper for the camera-ready.